# Implicit Bias of Projected Subgradient Method Gives Provable Robust Recovery of Subspaces of Unknown Codimension

**Paris Giampouras, Benjamin D. Haeffele and René Vidal**
Mathematical Institute for Data Science
Johns Hopkins University
Baltimore, MD, USA
`{parisg,bhaeffele,rvidal}@jhu.edu`

## Abstract

Robust subspace recovery (RSR) is the problem of learning a subspace from sample data points corrupted by outliers. Dual Principal Component Pursuit (DPCP) is a robust subspace recovery method that aims to find a basis for the orthogonal complement of the subspace by minimizing the sum of the distances of the points to the subspaces subject to orthogonality constraints on the basis. Prior work has shown that DPCP can provably recover the correct subspace in the presence of outliers as long as the true dimension of the subspace is known. In this paper, we show that if the orthogonality constraints –adopted in previous DPCP formulations– are relaxed and random initialization is used instead of spectral one, DPCP can provably recover a subspace of *unknown dimension*. Specifically, we propose a very simple algorithm based on running multiple instances of a projected sub-gradient descent method (PSGM), with each problem instance seeking to find one vector in the null space of the subspace. We theoretically prove that under mild conditions this approach succeeds with high probability. In particular, we show that 1) all of the problem instances will converge to a vector in the nullspace of the subspace and 2) the ensemble of problem instance solutions will be sufficiently diverse to fully span the nullspace of the subspace thus also revealing its true unknown codimension. We provide empirical results that corroborate our theoretical results and showcase the remarkable implicit rank regularization behavior of the PSGM algorithm that allows us to perform RSR without knowing the subspace dimension.

## 1 Introduction

Robust subspace recovery (RSR) refers to the problem of identifying an underlying linear subspace (with dimension less than the ambient data dimension) from sample data points that are potentially corrupted with outliers (i.e., points that do not lie in the linear subspace). Many methods for RSR have been proposed in the literature over the past several years (Xu et al., 2012; You et al., 2017a; Lerman & Maunu, 2018). Formulations based on convex relaxations and decompositions of the data matrix into a low-rank matrix plus a matrix of sparse corruptions – either entrywise-sparse corruptions as in Candès et al. (2011) or columnwise-sparse corruptions as in Xu et al. (2012); McCoy & Tropp (2011) – can, in certain situations, be shown to provably recover the true subspace when the dimension is unknown. However, these theoretical guarantees often require the dimension of the subspace, $d$, to be significantly less than the ambient dimension of the data, $D$, and these methods are not suitable for the more challenging regime of subspaces of high relative dimension (i.e., when $\frac{d}{D} \approx 1$).

**Dual Principal Component Pursuit.** Recently, progress has been made towards solving the RSR problem in the high relative dimension regime by a formulation termed Dual Principal Component Pursuit (DPCP). As implied by its name, DPCP follows a *dual* perspective of RSR by aiming to recover a basis for the orthogonal complement of the inliers' subspace. As shown in (Tsakiris & Vidal, 2018), DPCP is provably robust in recovering subspaces of high relative dimension. However, a key limitation of DPCP is that it requires *a priori* knowledge of the true subspace dimension.

**DPCP for** $c = 1$. Let $\tilde{X} \in \mathbb{R}^{D \times (N+M)}$ denote the data matrix defined as $\tilde{X} = [X \ O]\Gamma$, where $X \in \mathbb{R}^{D \times N}$ is a matrix containing $N$ inliers as its columns, $O \in \mathbb{R}^{D \times M}$ is a matrix containing $M$ outliers, and $\Gamma$ is an unknown permutation matrix. DPCP was first formulated by Tsakiris & Vidal (2018) for handling subspaces of codimension $c = D - d$ equal to 1 (i.e., the subspace is a hyperplane with dimension $d = D - 1$). In this case, DPCP is formulated as the optimization problem

$$\min_{b \in \mathbf{R}^D} \ \|\tilde{X}^\top b\|_1 \ \text{ s.t. } \ \|b\|_2 = 1, \tag{1}$$

which is nonconvex due to the spherical constraint imposed on the normal vector $b \in \mathbb{S}^{D-1}$ of the $D - 1$ dimensional hyperplane. Tsakiris & Vidal (2018) showed that the global minimizer of (1) is a normal vector of the underlying true hyperplane when both inliers and outliers are well-distributed *or* the ratio between the number of inliers and number of outliers is sufficiently small. Following a probabilistic point of view, (Zhu et al., 2018) presented an improved theoretical analysis of DPCP giving further insights on the remarkable robustness of DPCP in recovering the true underlying subspaces even in datasets heavily corrupted by outliers. Moreover, the authors introduced a projected subgradient method which converges to a normal vector of the true subspace at a linear rate.

**Recursive DPCP for known** $c > 1$. Zhu et al. (2018) also proposed an extension to DPCP to subspaces with codimension $c > 1$ via a projected subgradient algorithm that tries to learn $c$ normal vectors to the subspace in a recursive manner. Specifically, after convergence to a normal vector, the projected subgradient algorithm is initialized with a vector *orthogonal* to the previously estimated normal vector. However, for that approach to be successful, knowledge of the true subspace codimension $c$ becomes critical. Specifically, if an underestimate of the true codimension $c$ is assumed the recovered basis for the null space, $\hat{B}$, will fail to span the whole null space, $\mathcal{S}_\perp$. On the other hand, an overestimate of $c$ will lead to columns of $\hat{B}$ corresponding to vectors that lie in $\mathcal{S}$.

**Orthogonal DPCP for known** $c$. Zhu et al. (2019) proposed an alternative to (1) which attempts to solve for $c$ normal vectors to the subspace at once by minimizing the sum of the distances from the points to the subspace:

$$\min_{B \in \mathbb{R}^{D \times c}} \|\tilde{X}^\top B\|_{1,2} \quad \text{s.t.} \quad B^\top B = I. \tag{2}$$

The authors also proposed an optimization algorithm based on the projected Riemannian subgradient method (RSGM), which builds on similar ideas as the projected subgradient method of Zhu et al. (2018) and enjoys a linear converge rate when the step size is selected based on a geometrically diminishing rule. Ding et al. (2021) provided a geometric analysis of (2) which shows the merits of DPCP in handling a) datasets highly contaminated by outliers (in the order of $M = \mathcal{O}(N^2)$) and b) subspaces of high relative dimension. However, a key shortcoming of this approach is that, because all minimizers of (2) are orthogonal matrices, a prerequisite for recovering the correct orthogonal complement of the inliers subspace is the *a priori* knowledge of the true codimension $c$ (see Fig. 1).

**Contributions.** In this work, we address this key limitation by proposing a framework that allows us to perform robust subspace recovery in the high relative subspace dimension regime **without** requiring *a priori* knowledge of the true subspace dimension. In particular, our proposed approach is based on the simple idea of solving multiple, parallel instances of the DPCP formulation for subspaces of codimension one,

$$\min_{B \in \mathbb{R}^{D \times c'}} \ \sum_{i=1}^{c'} \|\tilde{X}^\top b_i\|_1 \ \text{ s.t.} \quad \|b_i\|_2 = 1, \quad i = 1, 2, \ldots, c' \tag{3}$$

where $c'$ is assumed to be an upper bound of $c$, i.e., $c' \geq c$. Contrary to (2), the objective function in (3) decouples over the columns $b_i$ of matrix $B = [b_1 \ b_2 \ \cdots b_{c'}]$ and thus can be solved in a parallel manner by independently applying a projected subgradient algorithm (referred to as PSGM) from $c'$ different *random initializations*. Moreover, we observe that with random initialization we can get vectors sufficiently spread on the sphere that lead PSGM (initialized with those vectors) to return normal vectors of $\mathcal{S}$. These are all *linearly independent* when $c' \leq c$ and thus can span $\mathcal{S}_\perp$ when $c' = c$. If $c' > c$ then PSGM will return $c' - c$ redundant vectors that will still lie in $\mathcal{S}_\perp$ yet they will be linearly dependent (see Figure 1). That being said, we show that this simple strategy permits us to robustly recover the true subspace even without knowledge of the true codimension $c$.

As is detailed in Sections 3 and 4, this remarkable behavior of PSGM originates from the *implicit bias* that is induced in the optimization problem due to a) the relaxation of *orthogonality constraints* in (3) and b) the *random initialization* scheme that is adopted. Our specific contributions are as follows:

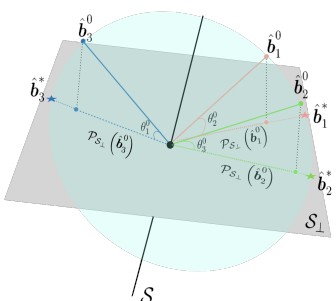 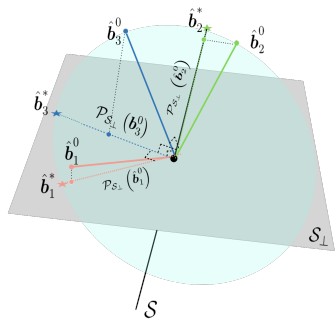

Figure 1: Graphical illustration of the recovered normal vectors of $\mathcal{S}$ by (left) the proposed DPCP-PSGM approach and (right) methods that use spectral initialization and impose orthogonality constraints . Initial vectors $\boldsymbol{b}_1^0, \boldsymbol{b}_2^0, \boldsymbol{b}_3^0$ are randomly initialized and are non-orthogonal in (left) and spectrally initialized (hence orthogonal) in (right). Note that in (left) $\mathrm{rank}(\hat{\boldsymbol{B}}^*)$ (where $\hat{\boldsymbol{B}}^* = [\boldsymbol{b}_1^*, \boldsymbol{b}_2^*, \boldsymbol{b}_3^*]$) equals to the true codimension $c = 2$ of $\mathcal{S}$ and $\mathrm{span}(\boldsymbol{B}^*) \equiv \mathcal{S}_\perp$ while in (right) $\boldsymbol{B}^*$ is orthogonal hence $\mathrm{rank}(\boldsymbol{B}^*) = 3$ with $\boldsymbol{b}_2^* \in \mathcal{S}$.

1. First, we study a continuous version of (3) where the inliers and outliers are drawn from continuous measures. We show that this induces a benign landscape on the DPCP objective, which can be analyzed more easily. Specifically, we prove that the DPCP problem in (3) can be solved via a projected subgradient algorithm that implicitly biases solutions towards low-rank matrices $\hat{\boldsymbol{B}} \in \mathbb{R}^{D \times c'}$ whose columns are the projections of the randomly initialized columns of $\boldsymbol{B}^0$ onto $\mathcal{S}_\perp$. As a result, $\hat{\boldsymbol{B}}$ *almost surely* spans $\mathcal{S}_\perp$ as long as it is *randomly initialized* with $c' \geq c$.

2. Second, we analyze the discrete version which is more challenging, yet of more practical interest, showing that iterates of DPCP-PSGM converge to a *scaled* and *perturbed* version of the initial matrix $\boldsymbol{B}^0$. This compelling feature of DPCP-PSGM allows to derive a sufficient condition and a probabilistic bound guaranteeing when the matrix $\hat{\boldsymbol{B}} \in \mathbb{R}^{D \times c'}$ spans $\mathcal{S}_\perp$.

3. We provide empirical results both on simulated and a real datasets, corroborating our theory and showing the robustness of our approach even without knowledge of the true subspace codimension.

## 2 RELATED WORK

**Subspace Recovery.** Learning underlying low-dimensional subspace representations of data has been a central topic of interest in machine learning research. Principal Component Analysis (PCA) has been the most celebrated method of this kind and is based on the minimization of the perpendicular distances of the data points from the estimates linear subspace, Jolliffe & Cadima (2016). Albeit, it is originally formulated as nonconvex optimization problem, PCA can be easily solved in closed form using a singular value decomposition (SVD) operation (see e.g. Vidal et al. (2016)). Despite its great success, PCA is prone to failure when handling datasets that contain *outliers* i.e., data points whose deviation from the inliers' subspace is "large" in the $\ell_2$ norm sense.

**Robust Subspace Recovery (RSR).** To remedy this weakness of PCA *robust subspace recovery (RSR)* methods attempt to identify the outliers in the dataset an recover the true underlying low-dimensional subspace of the inliers Lerman & Maunu (2018); Maunu et al. (2019). A classical approach to this problem is RANSAC Fischler & Bolles (1981), which is given a time budget and randomly chooses per iteration $d$ points and then fits a $d$-dimensional subspace to those points and checks how the proposed subspace fits the remaining data points. RANSAC then outputs the subspace that agrees with the largest number of points. However, RANSAC's reliance on randomly sampling points to propose subspaces can be highly inefficient when the number of outliers is high (as well as the fact that RANSAC also needs knowledge of the true subspace dimension, $d$). The need to tackle inherent shortcomings of RANSAC pertaining to computational complexity issues inspired alternative convex formulations of RSR, Xu et al. (2012); You et al. (2017b); Rahmani & Atia (2017); Zhang & Lerman (2014). In Xu et al. (2012) the authors decompose the data matrix as a sum of a low-rank and a column-sparse component. However, theoretical guarantees obtained for convex

formulations only hold for subspaces of relatively low-dimensional subspaces i.e., for $d \ll D$ where $d$ and $D$ denote the subspace and the ambient dimension, respectively. To the best of our knowledge, existing RSR algorithms rely heavily on one of two key assumptions. 1) The subspace is very low-dimensional relative to the ambient dimension ($d \ll D$) or 2) *The subspace dimension is a priori known*. Undoubtedly, the second hypothesis is rather strong in real-world applications, and many applications also do not satisfy the first assumption. Moreover, heuristic strategies for selecting the dimension of the subspace are hard to be applied in the RSR setting since they incur computationally prohibitive procedures, Lerman & Maunu (2018).

**Relation to Orthogonal Dictionary Learning (ODL).** Note that objective functions in the form of (3) show up beyond RSR problems i.e., in orthogonal dictionary learning (ODL), sparse blind deconvolution, etc., Qu et al. (2020). Specifically, based on a similar formulation the authors in Bai et al. (2019) proved that $c' = \mathcal{O}(c \log c)$ independent random initial vectors suffice in order to recover with high probability a dictionary of size $D \times c$ with high accuracy. In this paper we aim to recover a basis of the orthogonal complement of a subspace of unknown dimension instead of accurately estimating a dictionary hence our goal differs from that in Bai et al. (2019).

**Implicit bias in Robust Recovery Problems.** The notions of implicit bias and implicit regularization have been used interchangeably in the nonconvex optimization literature for describing the tendency of optimization algorithms to converge to global minima of minimal complexity with favorable generalization properties in overparameterized models, Gunasekar et al. (2018). In the context of robust recovery, the authors in You et al. (2020) showed that Robust PCA can be suitably re-parametrized in such a way to favor low-rank and sparse solutions without using any explicit regularization. In this work, we use the term implicit bias for describing the convergence of DPCP-PSGM to low-rank solutions, which are not necessarily global minimizers, that span the orthogonal complement of the subspace when a) orthogonality constraints in DPCP formulation are relaxed b) DPCP is overparameterized i.e., $c' \geq c$ and c) PSGM randomly initialized.

## 3 DUAL PRINCIPAL COMPONENT PURSUIT AND THE PROJECTED SUBGRADIENT METHOD

We re-write the DPCP formulation given in (1) as

$$\min_{\boldsymbol{B} \in \mathbb{R}^{D \times c'}} \|\tilde{\boldsymbol{X}}^\top \boldsymbol{b}\|_1 = \|\boldsymbol{X}^\top \boldsymbol{b}\|_1 + \|\boldsymbol{O}^\top \boldsymbol{b}\|_1 \quad \text{s.t.} \quad \|\boldsymbol{b}\|_2 = 1 \tag{4}$$

In Zhu et al. (2018), the authors proposed a projected subgradient descent algorithm for addressing (4) that consists of a subgradient step followed by a projection onto the sphere i.e.,

$$\boldsymbol{b}^{k+1} = \hat{\boldsymbol{b}}^k - \mu^k \left( \boldsymbol{X} \text{Sgn}(\boldsymbol{X}^\top \boldsymbol{b}^k) + \boldsymbol{O} \text{Sgn}(\boldsymbol{O}^\top \boldsymbol{b}^k) \right) \quad \text{and} \quad \hat{\boldsymbol{b}}^{k+1} = \mathcal{P}_{\mathbb{S}^{D-1}}(\boldsymbol{b}^{k+1}), \tag{5}$$

where $\mu^k$ is the -adaptively updated per iteration- step size and $\hat{\boldsymbol{b}}^k$ is the unit $\ell_2$ norm vector corresponding to the $k$th iteration.

The convergence properties of the projected subgradient algorithm described above depend on specific quantities denoted as $c_{\boldsymbol{X},\min}$ and $c_{\boldsymbol{X},\max}$ that reflect the geometry of the problem and are defined as $c_{\boldsymbol{X},\min} = \frac{1}{N} \min_{\boldsymbol{b} \in \mathbb{S}^{d-1} \cap \mathcal{S}} \|\boldsymbol{X}^\top \boldsymbol{b}\|_1$ and $c_{\boldsymbol{X},\max} = \frac{1}{N} \max_{\boldsymbol{b} \in \mathbb{S}^{D-1} \cap \mathcal{S}} \|\boldsymbol{X}^\top \boldsymbol{b}\|_1$. Note that the more well distributed the inliers are in the subspace $\mathcal{S}$ the higher the value of the quantity $c_{\boldsymbol{X},\min}$ (called as permeance statistic which first appeared in Lerman et al. (2015)) as it becomes harder to find a vector $\boldsymbol{b}$ in the subspace $\mathcal{S}$ that is orthogonal to the inliers. Moreover, $c_{\boldsymbol{X},\min}$ and $c_{\boldsymbol{X},\max}$ converge to the same value as $N \to \infty$ provided the inliers are uniformly distributed in the subspace i.e., $c_{\boldsymbol{X},\min} \to c_d, c_{\boldsymbol{X},\max} \to c_d$, where $c_d$ is given as the average height of the unit hemisphere on $\mathbb{R}^d$,

$$c_d := \frac{(d-2)!!}{(d-1)!!} \begin{cases} \frac{2}{\pi}, & \text{if } d \text{ is even,} \\ 1, & \text{if } d \text{ is odd} \end{cases} \quad \text{where } k!! = \begin{cases} k(k-2)(k-4) \cdots 4 \cdot 2, & k \text{ is even,} \\ k(k-2)(k-4) \cdots 3 \cdot 1, & k \text{ is odd} \end{cases} \tag{6}$$

Similarly to $c_{\boldsymbol{X},\min}, c_{\boldsymbol{X},\max}$, we will also be interested in quantities $c_{\boldsymbol{O},\min}, c_{\boldsymbol{O},\max}$ which indicate how well-distributed the outliers are in the ambient space. These quantities are defined as $c_{\boldsymbol{O},\min} = \min_{\boldsymbol{b} \in \mathbb{S}^{D-1}} \frac{1}{M} \|\boldsymbol{O}^\top \boldsymbol{b}\|_1$ and $c_{\boldsymbol{O},\max} = \max_{\boldsymbol{b} \in \mathbb{S}^{D-1}} \frac{1}{M} \|\boldsymbol{O}^\top \boldsymbol{b}\|_1$. $c_{\boldsymbol{O},\max}$ can be viewed as the *dual* permeance statistic and is bounded away from small values while its difference from $c_{\boldsymbol{O},\min}$ tends to

zero as $M \to \infty$. Further, if the outliers are uniformly distributed on the sphere, then $c_{\boldsymbol{O},\max} \to c_D$ and $c_{\boldsymbol{O},\min} \to c_D$ where $c_D$ is defined as in (6), (Zhu et al., 2018).

Finally, we also define the quantities $\eta_{\boldsymbol{O}} = \frac{1}{M} \max_{\boldsymbol{b} \in \mathbb{S}^{D-1}} \|(\boldsymbol{I} - \boldsymbol{b}\boldsymbol{b}^\top)\boldsymbol{O}\mathrm{Sgn}(\boldsymbol{O}^\top \boldsymbol{b})\|_2$ and $\eta_{\boldsymbol{X}} = \frac{1}{M} \max_{\boldsymbol{b} \in \mathbb{S}^{D-1}} \|(\mathcal{P}_{\mathcal{S}} - \boldsymbol{b}\boldsymbol{b}^\top)\boldsymbol{X}\mathrm{Sgn}(\boldsymbol{X}^\top \boldsymbol{b})\|_2$. As $M \to \infty$ and assuming outliers in $\boldsymbol{O}$ are well-distributed we get $\boldsymbol{O}\mathrm{Sgn}(\boldsymbol{O}^\top \boldsymbol{b}) \to c_D \boldsymbol{b}$ thus $\eta_{\boldsymbol{O}} \to 0$ (Tsakiris & Vidal, 2018). Likewise, $\eta_{\boldsymbol{X}} \to 0$ as $N \to \infty$ provided that inliers are uniformly distributed in the $d$-dimensional subspace. The following theorem (see full version in Appendix) provides convergence guarantees of the projected subgradient method that was proposed in Zhu et al. (2018) for addressing problem (1).

**Theorem 1** *(Informal Theorem 3 of Zhu et al. (2018)) Let $\{\hat{\boldsymbol{b}}_k\}$ the sequence generated by the projected subgradient algorithm in Zhu et al. (2018), with initialization $\hat{\boldsymbol{b}}_0$ such that*

$$\theta_0 < \arctan\left(\frac{Nc_{\boldsymbol{X},\min}}{N\eta_{\boldsymbol{X}} + M\eta_{\boldsymbol{O}}}\right) \ \text{ and } \ Nc_{\boldsymbol{X},\min} \geq N\eta_{\boldsymbol{X}} + M\eta_{\boldsymbol{O}} \tag{7}$$

*where $\theta_0$ denotes the principal angle of $\boldsymbol{b}^0$ from $\mathcal{S}_\perp$. If the step size $\mu^k$ is updated according to a piecewise geometrically diminishing rule given as*

$$\mu^k = \begin{cases} \mu^0, & k < K_0 \\ \mu^0 \beta^{\lfloor (k-K_0)/K_* \rfloor + 1}, & k \geq K_0 \end{cases} \tag{8}$$

*where $\beta < 1$, $\lfloor \cdot \rfloor$ is the floor function, then the iterates $\boldsymbol{b}^k$ converge to a normal vector of $\mathcal{S}$.*

## 4 DUAL PRINCIPAL COMPONENT PURSUIT IN SUBSPACES OF UNKNOWN CODIMENSION

Current theoretical results provide guarantees for recovering the true inlier subspace, when the proposed algorithms know *a priori* of the subspace codimension $c$, which is a rather strong requirement and is far from being true in real word applications. Here we describe our proposed approach, which consists of removing the orthogonality constraint on $\boldsymbol{B}$, along with a theoretical analysis that gives guarantees of recovering the true underlying subspace even when the true codimension $c$ is unknown. First we analyze a continuous version of DPCP, which arises when the number of inliers and outliers are distributed according to continuous measures and their number tends to $\infty$. The continuous DPCP incurs an optimization problem with a benign landscape that allows us to better illustrate the favorable properties of DPCP-PSGM when it comes to the convergence of its iterates. Then we extend the results to the discrete case that deals with a finite number of inliers and outliers yielding a more challenging optimization landscape.

### 4.1 PSGM'S ITERATES CONVERGENCE IN THE CONTINUOUS VERSION OF DPCP

The following lemma provides the continuous version of the discrete objective function given in (3).

**Lemma 2** *In the continuous case, the discrete DPCP problem given in (3) is reformulated as,*

$$\min_{\boldsymbol{B} \in \mathbb{R}^{D \times c'}} \sum_{i=1}^{c'} \left( p\mathbb{E}_{\boldsymbol{\mu}_{\mathbb{S}^{D-1}}}[f_{\boldsymbol{b}_i}] + (1-p)\mathbb{E}_{\boldsymbol{\mu}_{\mathbb{S}^{D-1} \cap \mathcal{S}}}[f_{\boldsymbol{b}_i}] \right) = \sum_{i=1}^{c'} \|\boldsymbol{b}_i\|_2 \left( pc_D + (1-p)c_d\cos(\phi_i) \right)$$

$$\text{s.t. } \|\boldsymbol{b}_i\|_2 = 1, \ i = 1, 2, \ldots, c' \tag{9}$$

*where $f_{\boldsymbol{b}} : \mathbb{S}^{D-1} \to \mathbb{R}$, $f_{\boldsymbol{b}}(\mathbf{z}) = |\mathbf{z}^\top \boldsymbol{b}|$, $\phi_i$ is the principal angle of $\boldsymbol{b}_i$ from the inliers subspace $\mathcal{S}$ and $p$ is the probability of occurrence of an outlier.*

Note that $\boldsymbol{\mu}_{\mathbb{S}^{D-1}}, \boldsymbol{\mu}_{\mathbb{S}^{D-1} \cap \boldsymbol{S}}$ are the continuous measures associated with the outliers and inliers, respectively. Evidently, (9) attains its global minimum for vectors $\boldsymbol{b}_i$s that are orthogonal to the inliers' subspace. Based on (3) and due to Lemma 2, we can now minimize the objective function of the "continuous version" of DPCP by employing a projected subgradient methods (PSGM) that

performs the following steps per iteration [1]

$$\boldsymbol{b}_i^{k+1} = \hat{\boldsymbol{b}}_i^k - \mu_i^k(pc_D\hat{\boldsymbol{b}}_i^k + (1-p)c_d\hat{\boldsymbol{s}}_i^k) \text{ and } \hat{\boldsymbol{b}}_i^{k+1} = \mathcal{P}_{\mathcal{S}_\perp}(\boldsymbol{b}_i^{k+1}), i = 1, 2, \ldots, c' \tag{10}$$

**Lemma 3** *A projected subgradient algorithm consisting of the steps described in (10) using a piecewise geometrically diminishing step size rule (see (8) in Theorem 1) will almost surely asymptotically converge to a matrix $\hat{\boldsymbol{B}}^* \in \mathbb{R}^{D \times c'}$ whose columns $\hat{\boldsymbol{b}}_i^*$, $i = 1, 2, \ldots, c'$ will be normal vectors of the inliers' subspace when randomly initialized with vectors $\boldsymbol{b}_i^0 \in \mathbb{S}^{D-1}$, $i = 1, 2, \ldots, c'$ uniformly distributed over the sphere $\mathbb{S}^{D-1}$.*

Lemma 3 allows us to claim that we can always recover $c' \geq c$ normal vectors to the inliers' subspace using a PSGM algorithm consisting of steps given in (10). However, this does not tell the whole story yet, since our ultimate objective is to recover a matrix $\hat{\boldsymbol{B}}$ that spans $\mathcal{S}_\perp$. Thus, it remains to show that the rank of $\hat{\boldsymbol{B}}$ is equal to the true and *unknown* codimension of the inliers' subspace $c$. Next we prove that by initializing with a $\hat{\boldsymbol{B}}_0$ such that $\text{rank}(\hat{\boldsymbol{B}}_0) = c'$ (i.e., $\hat{\boldsymbol{B}}_0$ is initialized to be full-rank), we can guarantee that we can solve the continuous version of DPCP using PSGM and converge to a $\hat{\boldsymbol{B}}$ such that $\text{rank}(\hat{\boldsymbol{B}}) = c$ thus getting $\text{span}(\hat{\boldsymbol{B}}) \equiv \mathcal{S}_\perp$ (along with recovering the true subspace dimension). By projecting the PSGM iterates given in (10) onto $\mathcal{S}_\perp$ we have,

$$\mathcal{P}_{\mathcal{S}_\perp}(\boldsymbol{b}_i^{k+1}) = (1 - \mu_i^k pc_D)\mathcal{P}_{\mathcal{S}_\perp}(\hat{\boldsymbol{b}}_i^k) \text{ and } \mathcal{P}_{\mathcal{S}_\perp}(\hat{\boldsymbol{b}}_i^{k+1}) = \mathcal{P}_{\mathcal{S}_\perp}(\mathcal{P}_{\mathbb{S}^{D-1}}(\boldsymbol{b}_i^{k+1})) \tag{11}$$

We hence observe that the projections of successive iterates of PSGM are scaled versions of the corresponding projections of the previous iterates. We can now state Lemma 4.

**Lemma 4** *The PSGM iterates $\hat{\boldsymbol{b}}_i^k$, $i = 1, 2, \ldots, c'$, $k = 1, 2, \ldots$ given in (10), when randomly initialized with $\hat{\boldsymbol{b}}_i^0 s$, $i = 1, 2, \ldots, c'$ that are independently drawn from a spherical distribution with unit $\ell_2$ norm converge almost surely to $c'$ normal vectors of the inliers subspace $\mathcal{S}$ denoted as $\hat{\boldsymbol{b}}_i^*$, $i = 1, 2, \ldots, c'$ that are given by $\hat{\boldsymbol{b}}_i^* = \frac{\mathcal{P}_{\mathcal{S}_\perp}(\hat{\boldsymbol{b}}_i^0)}{\|\mathcal{P}_{\mathcal{S}_\perp}(\hat{\boldsymbol{b}}_i^0)\|_2}$, $i = 1, 2, \ldots, c'$.*

Lemma 4 shows that the initialization of PSGM plays a pivotal role since it determines the direction of the recovered normal vectors $\{\hat{\boldsymbol{b}}_i^*\}_{i=1}^{c'}$. Lemmas 3 and 4 pave the way for Theorem 5.

**Theorem 5** *Let $\hat{\boldsymbol{B}}^0 \in \mathbb{R}^{D \times c'}$ where $c' \geq c$ with $c$ denoting the true codimension of the inliers subspace $\mathcal{S}$, consisting of unit $\ell_2$ norm column vectors $\hat{\boldsymbol{b}}_i^0 \in \mathbb{S}^{D-1}, i = 1, 2, \ldots, c'$ that are independently drawn from uniform distribution over the sphere $\mathbb{S}^{D-1}$. A PSGM algorithm initialized with $\hat{\boldsymbol{B}}^0$ will almost surely converge to a matrix $\hat{\boldsymbol{B}}^*$ such that $\text{span}(\hat{\boldsymbol{B}}^*) \equiv \mathcal{S}_\perp$.*

From Theorem 5 we observe that in the benign scenario where inliers and outliers are distributed under continuous measures, we can recover the correct orthogonal complement of the inlier's subspace even when we are oblivious to its true codimension. Remarkably, this is achieved by exploiting the *implicit bias* induced by multiple random initializations of the PSGM algorithm for solving the DPCP formulation given in (3), which is free of orthogonality constraints.

## 4.2 PSGM'S ITERATES CONVERGENCE IN THE DISCRETE VERSION OF DPCP

From this analysis of the continuous version of DPCP we now extend to the the discrete version, which is of more practical relevance for finite data, yet also presents more challenges. To begin, we reformulate the DPCP objective as follows

$$\sum_{i=1}^{c'} \|\tilde{\boldsymbol{X}}^\top \boldsymbol{b}_i\|_1 = \sum_{i=1}^{c'} \|\boldsymbol{X}^\top \boldsymbol{b}_i\|_1 + \|\boldsymbol{O}^\top \boldsymbol{b}_i\|_1 = M \sum_{i=1}^{c'} \boldsymbol{b}_i^\top \boldsymbol{o}_{\boldsymbol{b}_i} + N \sum_{i=1}^{c'} \boldsymbol{b}_i^\top \boldsymbol{x}_{\boldsymbol{b}_i} \tag{12}$$

where $\boldsymbol{x}_{\boldsymbol{b}_i}$ and $o_{\boldsymbol{b}_i}$ are called as *average inliers* and *average outliers* terms, defined as $\boldsymbol{x}_{\boldsymbol{b}_i} = \frac{1}{N}\sum_{j=1}^{N} Sgn(\boldsymbol{b}_i^\top \boldsymbol{x}_j)\boldsymbol{x}_j$ and $\boldsymbol{o}_{\boldsymbol{b}_i} = \frac{1}{M}\sum_{j=1}^{M} Sgn(\boldsymbol{b}_i^\top \boldsymbol{o}_j)\boldsymbol{o}_j$.

In Algorithm 1, we give the projected subgradient method (DPCP-PSGM) applied on the DPCP problem given in (3).

---

[1]Note that $\partial\|\boldsymbol{b}\|_2 = \frac{\boldsymbol{b}}{\|\boldsymbol{b}\|_2}$ for $\boldsymbol{b} \neq \boldsymbol{0}$ and $\|\boldsymbol{b}_i\|_2\cos(\phi_i) = \boldsymbol{b}_i^\top \hat{\boldsymbol{s}}_i$ where $\hat{\boldsymbol{s}}_i = \frac{\mathcal{P}_{\mathcal{S}}(\boldsymbol{b}_i)}{\|\mathcal{P}_{\mathcal{S}}(\boldsymbol{b}_i)\|_2}$.

---

**Algorithm 1:** DPCP-PSGM algorithm for solving (3)

**Result:** $\hat{B} = [\hat{b}_1^k, \hat{b}_2^k, \ldots, \hat{b}_{c'}^k]$

**Initialize:** Randomly sample $\hat{b}_0^1, \hat{b}_0^2, \ldots, \hat{b}_0^{c'}$ from a uniform distribution on $\mathbb{S}^{D-1}$ ;

**for** $k = 1, 2, \ldots$ **do**

    **for** $i = 1, 2, \ldots, c'$ **do**

        Update the step-size according to a specific rule;

        $b_i^{k+1} = \hat{b}_i^k - \mu_i^k (M o_{\hat{b}_i}^k + N x_{\hat{b}_i}^k)$;

        $\hat{b}_i^{k+1} = \mathcal{P}_{\mathbb{S}^{D-1}}(b_i^{k+1})$;

    **end**

**end**

---

Note that the average outliers and inliers terms are discrete versions of the corresponding continuous average terms $c_D b_i$ and $c_d \hat{s}_i$ where $\hat{s}_i = \mathcal{P}_{\mathcal{S}}(b_i)$, respectively, Tsakiris & Vidal (2018). We now express the sub-gradient step of Algorithm 1 as

$$b_i^{k+1} = \hat{b}_i^k - \mu_i^k \left( M(c_D \hat{b}_i^k + e_O^{i,k}) + N(c_D \hat{s}_i^k + e_X^{i,k}) \right) \tag{13}$$

where the quantities $e_O^{i,k} = o_{\hat{b}_i^k} - c_D \hat{b}_i^k -$ and $e_X^{i,k} = x_{\hat{b}_i^k} - c_D \hat{s}_i^k$ account for the error between the continuous and discrete versions of the average outliers and the average inliers terms, respectively. Following a similar path as in the continuous case we next project the iterates of (13) onto $\mathcal{S}_\perp$,

$$\mathcal{P}_{\mathcal{S}_\perp}(b_i^{k+1}) = \mathcal{P}_{\mathcal{S}_\perp}(\hat{b}_i^k) - \mu_i^k \left( M \mathcal{P}_{\mathcal{S}_\perp}(c_D \hat{b}_i^k + e_O^{i,k}) + N \overset{0}{\cancel{\mathcal{P}_{\mathcal{S}_\perp}(c_D \hat{s}_i^k + e_X^{i,k})}} \right)$$

$$= (1 - \mu_i^k M c_D)\mathcal{P}_{\mathcal{S}_\perp}(\hat{b}_i^k) - \mu_i^k M \mathcal{P}_{\mathcal{S}_\perp}(e_O^{i,k}) \tag{14}$$

**Remark.** *Eq. (14) reveals that DPCP-PSGM, applied on the discrete problem, gives rise to updates whose projections to $\mathcal{S}_\perp$ are* **scaled** *and* **perturbed** *versions of the previous estimates. The magnitude of perturbation depends on the discrepancy between the continuous and the discrete problem.*

Recall that $\mathcal{P}_{\mathcal{S}_\perp}(\hat{b}_i^k) = \frac{\mathcal{P}_{\mathcal{S}_\perp}(b_i^k)}{\|b_i^k\|_2}$, and we can rewrite the update of the 2nd iteration of DPCP-PSGM,

$$\mathcal{P}_{\mathcal{S}_\perp}(b_i^2) = \frac{(1 - \mu_i^1 M c_D)}{\|b_i^1\|_2} \left( (1 - \mu_i^0 M c_D)\mathcal{P}_{\mathcal{S}_\perp}(\hat{b}_i^0) - \mu_i^0 M \mathcal{P}_{\mathcal{S}_\perp}(e_O^{i,0}) \right) - \mu_i^1 M \mathcal{P}_{\mathcal{S}_\perp}(e_O^{i,1})$$

$$= \frac{(1 - \mu_i^1 M c_D)(1 - \mu_i^0 M c_D)}{\|b_i^1\|_2 \|b_i^0\|_2}\mathcal{P}_{\mathcal{S}_\perp}(\hat{b}_i^0) - \frac{(1 - \mu_i^1 M c_D)}{\|b_i^1\|_2}\mu_i^0 M \mathcal{P}_{\mathcal{S}_\perp}(e_O^{i,0}) - \mu_i^1 M \mathcal{P}_{\mathcal{S}_\perp}(e_O^{i,1}) \tag{15}$$

where we have assumed that $\|b_i^0\|_2 = 1$. By repeatedly applying the same steps, we can reach to the following recursive expression for $\mathcal{P}_{\mathcal{S}_\perp}(b_i^K)$,

$$\mathcal{P}_{\mathcal{S}_\perp}(b_i^K) = \left( \prod_{k=0}^{K-1} \frac{(1 - \mu_i^k M c_D)}{\|b_i^k\|_2} \right) \mathcal{P}_{\mathcal{S}_\perp}(b_i^0) - \sum_{k=0}^{K-1} \left( \prod_{j=k+1}^{K-1} \frac{(1 - \mu_i^j M c_D)}{\|b_i^j\|_2} \right) \mu_i^k M \mathcal{P}_{\mathcal{S}_\perp}(e_O^{i,k}) \tag{16}$$

where for $j > K - 1$ we set $\prod_{j=k+1}^{K-1} \frac{(1 - \mu_i^j M c_D)}{\|b_i^j\|_2} = 1$.

By dividing (16) with $\prod_{k=0}^{K-1} \frac{(1 - \mu_i^k M c_D)}{\|b_i^k\|_2}$ and by projecting onto the sphere $\mathbb{S}^{D-1}$ we get

$$\mathcal{P}_{\mathbb{S}^{D-1}}(\mathcal{P}_{\mathcal{S}_\perp}(b_i^K)) = \mathcal{P}_{\mathbb{S}^{D-1}}(\mathcal{P}_{\mathcal{S}_\perp}(b_i^0) - \mathcal{P}_{\mathcal{S}_\perp}(\delta_i^K)) \tag{17}$$

where $\delta_i$ is defined as $\delta_i^K = \sum_{k=0}^{K-1} \left( \prod_{j=0}^{k} \frac{\|b_i^j\|_2}{(1 - \mu_i^j M c_D)} \right) \mu_i^k M e_O^{i,k}$.

**Assumption 1.** *We assume that the principal angles $\theta_0^i$ for all $b_i^0$s satisfy the inequality $\theta_0^i < \arctan\left( \frac{N c_{X,\min}}{N \eta_X + M \eta_O} \right)$ $\forall i, i = 1, 2, \ldots, c'$.*

Assumption 1 essentially assumes that the sufficient condition given in eq. (7) required by PSGM algorithm for converging to a normal vector is satisfied which is the same condition for success in Zhu et al. (2018). Under Assumption 1 we can invoke the convergence properties of PSGM given in Theorem 1 and get as $K \to \infty, \mathcal{P}_{\mathbb{S}^{D-1}}(\mathcal{P}_{\mathcal{S}_\perp}(\boldsymbol{b}^K)) \to \hat{\boldsymbol{b}}^* \in \mathcal{S}_\perp \cap \mathbb{S}^{D-1}$. That being said, we denote $\hat{\boldsymbol{b}}_i^* = \mathcal{P}_{\mathbb{S}^{D-1}}\left(\mathcal{P}_{\mathcal{S}_\perp}(\boldsymbol{b}_i^0) - \mathcal{P}_{\mathcal{S}_\perp}(\hat{\boldsymbol{\delta}}_i)\right)$, where $\hat{\boldsymbol{\delta}}_i = \lim_{K \to \infty} \boldsymbol{\delta}_i^{K2}$. Following the same steps as in Section 4.1 and by defining matrices $\boldsymbol{B}^* = [\boldsymbol{b}_1^*, \boldsymbol{b}_2^*, \ldots, \boldsymbol{b}_{c'}^*], \boldsymbol{B}^0 = [\boldsymbol{b}_1^0, \boldsymbol{b}_2^0, \ldots, \boldsymbol{b}_{c'}^0], \hat{\boldsymbol{\Delta}} = [\hat{\boldsymbol{\delta}}_1, \hat{\boldsymbol{\delta}}_2, \ldots, \hat{\boldsymbol{\delta}}_{c'}]$ we can express the matrix $\boldsymbol{B}^*$ as $\hat{\boldsymbol{B}}^* = \mathcal{P}_{\mathbb{S}^{D-1}}\left(\mathcal{P}_{\mathcal{S}_\perp}\left(\boldsymbol{B}^0 - \hat{\boldsymbol{\Delta}}\right)\right)$ where $\boldsymbol{B}^*$ will now consist of normal vectors of the inliers' subspace. In order to guarantee that $\text{span}(\boldsymbol{B}^*) \equiv \mathcal{S}_\perp$ it thus suffices to ensure that $\text{rank}\,(\boldsymbol{B}^*) = c$. Here we show that a sufficient condition for this to hold is that the matrix $\boldsymbol{A} = \boldsymbol{B}^0 - \hat{\boldsymbol{\Delta}}$ is full-rank.

**Lemma 6** *If $\sigma_{c'}(\boldsymbol{B}^0) > \|\hat{\boldsymbol{\Delta}}\|_2$ then matrix $\boldsymbol{A} = \boldsymbol{B}^0 - \hat{\boldsymbol{\Delta}}$ is full-rank.*

From Lemma 6 we can see that the success of DPCP-PSGM hinges on how well-conditioned the matrix $\boldsymbol{B}^0$ is. Specifically, it says that if a lower-bound on the smallest singular is satisfied then DPCP-PSGM is guaranteed to converge to the correct complement of the inlier without knowledge of the correct codimension $c$. From this, we can prove the following Theorem.

**Theorem 7** *Let $\mathbf{B}^0 \in \mathbb{R}^{D \times c'}$ with columns randomly sampled from a unit $\ell_2$ norm spherical distribution where $c' \geq c$ with $c$ denoting the true codimension of the inliers subspace $\mathcal{S}$ that satisfies Assumption 1. If*

$$1 - C_1 \sqrt{\frac{c'}{D}} - \frac{\epsilon}{\sqrt{D}} > \sqrt{c'} \kappa(\boldsymbol{\eta}_{\mathcal{O}} + c_{\mathcal{O},\max} - c_d) \tag{18}$$

*where $\kappa = \max_i \frac{M\mu_i^0}{\beta^{K_0/K_*}(1-r_i)}$ and $r_i = \frac{\left(1+\mu_i^0(N(\boldsymbol{\eta}_{\boldsymbol{\chi}}+c_{\boldsymbol{\chi},\max})+M(\boldsymbol{\eta}_{\mathcal{O}}+c_{\mathcal{O},\max}))\right)}{1-\mu_i^0 Mc_D} \beta^{1/K_*}$ then with probability at least $1-2\exp(-\epsilon^2 C_2)$ (where $C_1, C_2$ are absolute constants), Algorithm 1 with a piecewise geometrically diminishing step size rule will converge to a matrix $\hat{\boldsymbol{B}}^*$ such that $\text{span}(\hat{\boldsymbol{B}}^*) \equiv \mathcal{S}_\perp$.*

Note that quantities $\beta, K_*, K_0$ are used in the step-size update rule that is used as defined in (8) (See also full version of Theorem 1 in Appendix)). Theorem 7 shows that we can randomly initialize DPCP-PSGM, with a matrix $\hat{\boldsymbol{B}}^0$ whose number of columns $c'$ is an overestimate of the true codimension $c$ of the inliers' subspace and with columns sampled *independently* by a uniform distribution over the unit sphere and recover a matrix that will span the orthogonal complement of $\mathcal{S}$. The probability of success depends on the geometry of the problem since condition (18) is trivially satisfied (RHS of (18) tends to 0 since $\boldsymbol{\eta}_{\mathcal{O}} \to 0$ and $c_{\mathcal{O},\max} \to c_d$) in the continuous case which incurs a benign geometry. Moreover, the a less benign geometry would increase the value of $(\boldsymbol{\eta}_{\mathcal{O}} + c_{\mathcal{O},\max} - c_d)$ thus requiring a smaller initial codimension $c'$ that would lead to larger values the LHS of (18).

## 5 NUMERICAL SIMULATIONS

In this section we demonstrate the effectiveness of the proposed DPCP formulation and the derived DPCP-PSGM algorithm in recovering orthogonal complements of subspace of unknown codimension. We compare the proposed algorithm with previously developed methods i.e., DPCP-IRLS Tsakiris & Vidal (2018) and the Riemannian Subgradient Method (RSGM) Zhu et al. (2019). Recall that both DPCP-IRLS and RSGM address DPCP problem by enforcing orthogonality constraints, and thus both algorithms are quite sensitive if the true codimension of $\mathcal{S}$ is not known *a priori*. Further, they are both initialized using of spectral initialization i.e., $\hat{\boldsymbol{B}}^0 \in \mathbb{R}^{D \times c'}$ which contains the first $c'$ eigenvectors of matrix $\tilde{\boldsymbol{\mathcal{X}}}\tilde{\boldsymbol{\mathcal{X}}}^\top$ as its columns, as proposed in Tsakiris & Vidal (2018).

**Robustness to outliers in the unknown codimension regime.** In this experiment we set the dimension of the ambient space to $D = 200$. We randomly generate $N$ inliers uniformly distributed with unit $\ell_2$ norm in a $d = 195$ dimensional subspace (hence for its codimension we have $c = D - d = 5$). Following a similar process we generate $M$ outliers that live in the ambient space and are sampled

---

[2]For the sake of brevity we assume that the step size has been selected such that existence of the limit is guaranteed. We refer the reader to the proof of Lemma 8 for further details.

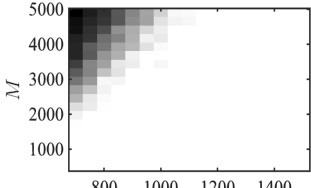 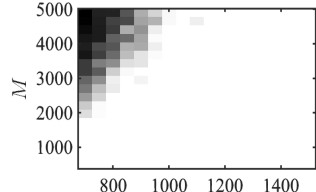 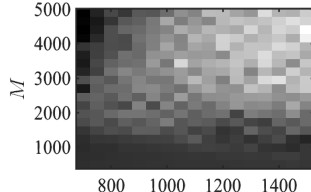

Figure 2: Distances of the recovered $\hat{\boldsymbol{B}}$ from the true orthogonal complements $\mathcal{S}_\perp$ as recovered by the proposed DPCP-PSGM algorithm provided an overestimated of the true $c$ i.e., $c' = 10$ (left), RSGM provided the true $c$ (middle) and RSGM provided $c' = 10$ (right). Darker colors reflect higher values of distances while lighter colors indicate successful recoveries of $\boldsymbol{B}$.

from a uniform distribution over the unit sphere. Fig. 2, illustrates the distances (see Appendix) of the recovered matrix $\hat{\boldsymbol{B}}$ as obtained by the proposed DPCP-PSGM algorithm initialized with an overestimate $c' = 10$ of the true codimension $c$ codimension and two versions of RSGM i.e., RSGM when it is given as input true $c = 5$ and RSGM when being incognizant of $c$ and hence it initialized with a $c' = 10$ of $c$As is shown in Fig. 2 (right), RSGM fails to recover the correct orthogonal complement of $\mathcal{S}$ when it is provided with an overestimate of the true $c$ which is attributed to spectral initialization and the imposed orthogonality constraints. On the contrary, DPCP-PSGM displays a remarkably robust behavior (Fig. 2(middle)) even without knowing the true value of $c$, performing similarly to RSGM when the latter knows beforehand the correct codimension (Fig. 2(left)).

**Recovery of the true codimension.** Here we test DPCP-PSGM on the recovery of the true codimension $c$ of the inliers' subspace $\mathcal{S}$. Again, we set $D = 200$ and generate $N = 1500$ inliers as before. We vary the true codimension of $\mathcal{S}$ from $c = 10$ to $20$ and consider two different outlier's ratios $r$, defined as $r = \frac{M}{M+N}$, namely $r = 0.6$ and $r = 0.7$. In both cases, DPCP-PSGM is initialized with the same overestimate of $c$ i.e., $c' = 30$. In Fig. 3 we report the estimated codimensions obtained by DPCP-PSGM for 10 independent trials of the experiments. It can be observed that DPCP-PSGM achieves 100% for all different codimensions for $r = 0.6$. Moreover, it shows a remarkable performance in estimating the correct $c$'s even in the more challenging case corresponding to outliers' ratios equal to 0.7. with the estimated codimensions being close to the true values even in the cases that it fails to exactly compute $c$. The results corroborate the theory showing that the DPCP-PSGM with random initialization biases the solutions of $\hat{\mathbf{B}}$ towards matrices with rank $c$.

## 6 CONCLUSIONS

We proposed a simple framework which allows us to perform robust subspace recovery without requiring a priori knowledge of the subspace codimension. This is based on Dual Principal Component Pursuit (DPCP) and thus is amenable to handling subspaces of high relative dimensions. We observed that a projected subgradient method

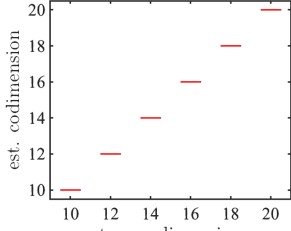 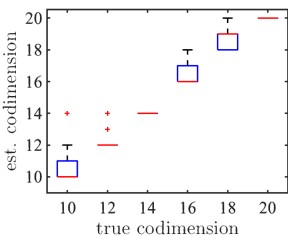

Figure 3: Estimated by DPCP-PSGM codimensions for two different outliers' ratios $r = \frac{M}{M+N}$ (a) $r = 0.6$ (left) and (b) $r = 0.7$ (right)

(PSGM) induces implicit bias and converges to a matrix that spans a basis of the orthogonal complement of the inliers subspace even as long as a) we overestimate it codimension, b) lift orthogonality constraints enforced in previous DPCP formulations and c) use random initialization. Empirical results that corroborate the developed theory and showcase the merits of our approach.

**Ethics Statement**   This work focuses on theoretical aspects of robust subspace recovery problem which is a well-established topic in machine learning research. The research conducted in the framework of this work raises no ethical issues or any violations vis-a-vis the ICLR Code of Ethics.

ACKNOWLEDGMENTS

We would like to thank Christian Kümmerle for helpful discussions on the probabilistic theorem that is used in Theorem 7. This work is partially supported by by the European Union under the Horizon 2020 Marie-Skłodowska- Curie Global Fellowship program: HyPPOCRATES— H2020-MSCA-IF-2018, Grant Agreement Number: 844290, and the NSF Grants 1704458, 2031985 and 1934979.

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

## A APPENDIX

**Theorem 1** *(Theorem 3 of Zhu et al. (2018)) Let $\{\hat{\boldsymbol{b}}_k\}$ the sequence generated by the projected subgradient method in Zhu et al. (2018), with initialization $\hat{\boldsymbol{b}}_0$ such that*

$$\theta_0 < \arctan\left(\frac{Nc_{\boldsymbol{X},\min}}{N\eta_{\boldsymbol{X}} + M\eta_{\boldsymbol{O}}}\right) \tag{19}$$

*where $\theta_0$ denotes the principal angle of $\boldsymbol{b}^0$ from $\mathcal{S}_\perp$, and*

$$Nc_{\boldsymbol{X},\min} \geq N\eta_{\boldsymbol{X}} + M\eta_{\boldsymbol{O}} \tag{20}$$

*Let $\mu' := \frac{1}{4\max\{Nc_{\boldsymbol{X},\min}, Mc_{\boldsymbol{O},\max}\}}$. If $\mu^0 \leq \mu'$ and the step size $\mu^k$ is updated according to a piece-wise geometrically diminishing rule given as*

$$\mu^k = \begin{cases} \mu^0, & k < K_0 \\ \mu^0\beta^{\lfloor(k-K_0)/K_*\rfloor+1}, & k \geq K_0 \end{cases} \tag{21}$$

*where $\beta < 1$, $\lfloor\cdot\rfloor$ is the floor function, and $K_0, K_* \in \mathbb{N}$ are chosen such that*

$$K_0 \geq K^\diamond(\mu^0),$$
$$K_* \geq \left(\sqrt{2}\beta\mu'\left(Nc_{\boldsymbol{X}} - (N\eta_{\boldsymbol{X}} + M\eta_{\boldsymbol{O}})\right)\right)^{-1}$$

*where,*

$$K^\diamond(\mu) := \frac{\tan(\theta_0)}{\mu\left(Nc_{\boldsymbol{X},\min} - \max\{1, \tan(\theta_0)\left(N\eta_{\boldsymbol{X}} + M\eta_{\boldsymbol{O}}\right)\}\right)} \tag{22}$$

*then for the angle $\theta_k$ between $\hat{\boldsymbol{b}}^k$ and $\mathcal{S}_\perp$ it holds*

$$\tan(\theta_0) \leq \begin{cases} \max\{\tan(\theta_0), \frac{\mu_0}{\sqrt{2}\mu'}\}, & k < K_0 \\ \frac{\mu^0}{\sqrt{2}\mu'}\beta^{\lfloor(k-K_0)/K_*\rfloor}, & k \geq K_0 \end{cases} \tag{23}$$

### A.1 PROOF OF LEMMA 2

**Lemma 2** *In the continuous case, the discrete DPCP problem given in (3) is reformulated as,*

$$\min_{\boldsymbol{B}\in\mathbb{R}^{D\times c'}} \sum_{i=1}^{c'} \left(p\mathbb{E}_{\boldsymbol{\mu}_{\mathbb{S}^{D-1}}}[f_{\boldsymbol{b}_i}] + (1-p)\mathbb{E}_{\boldsymbol{\mu}_{\mathbb{S}^{D-1}\cap\boldsymbol{S}}}[f_{\boldsymbol{b}_i}]\right) = \sum_{i=1}^{c'} \|\boldsymbol{b}_i\|_2 \left(pc_D + (1-p)c_d\cos(\phi_i)\right)$$
$$\text{s.t. } \|\boldsymbol{b}_i\|_2 = 1, \ i = 1, 2, \ldots, c' \tag{24}$$

*where $f_{\boldsymbol{b}} : \mathbb{S}^{D-1} \to \mathbb{R}$, $f_{\boldsymbol{b}}(\mathbf{z}) = |\mathbf{z}^\top\boldsymbol{b}|$, $\phi_i$ is the principal angle of $\boldsymbol{b}_i$ from the inliers subspace $\mathcal{S}$ and $p$ is the probability of occurrence of an outlier.*

**Proof** We define the discrete measures $\boldsymbol{\mu_{\mathcal{X}}}, \boldsymbol{\mu_{\mathcal{O}}}$ associated with the inliers and outliers, respectively as,

$$\boldsymbol{\mu_{\mathcal{X}}}(\mathbf{z}) = \frac{1}{N}\sum_{j=1}^{N} \delta(\mathbf{z} - \boldsymbol{o}_j), \ \ \boldsymbol{\mu_{\mathcal{O}}}(\mathbf{z}) = \frac{1}{M}\sum_{j}^{N} \delta(\mathbf{z} - \boldsymbol{x}_j) \tag{25}$$

where $\delta(\cdot)$ is the Dirac function. Recall that,

$$\int_{\mathbf{z}\in\mathbb{S}^{D-1}} g(\mathbf{z})\delta(\mathbf{z} - \mathbf{z}_0)d\boldsymbol{\mu}_{\mathbb{S}^{D-1}} = g(z_0) \tag{26}$$

where $g : \mathbb{S}^{D-1} \to \mathbb{R}$ and $\boldsymbol{\mu}_{\mathbb{S}^{D-1}}$ is the uniform measure on $\mathbb{S}^{D-1}$.

The DPCP objective for the discrete version of the problem divided by $M + N$ can be written as,

$$\frac{1}{M+N} \sum_{i=1}^{c'} \|\tilde{\boldsymbol{\mathcal{X}}}^\top \mathbf{b}_i\|_1 = \frac{1}{M+N} \sum_{i=1}^{c'} \left( \|\boldsymbol{\mathcal{X}}^\top \mathbf{b}_i\|_1 + \|\boldsymbol{\mathcal{O}}^\top \mathbf{b}_i\|_1 \right) = \frac{1}{M+N} \sum_{i}^{c'} \left( \sum_{j=1}^{N} |\boldsymbol{x}_j^\top \mathbf{b}_i| + \sum_{j=1}^{M} |\boldsymbol{o}_j^\top \mathbf{b}_i| \right)$$

$$= \frac{1}{M+N} \sum_{i=1}^{c'} \left( \sum_{j=1}^{N} \int_{\mathbf{z} \in \mathbb{S}^{D-1}} |\mathbf{z}^\top \mathbf{b}_i| \delta(\mathbf{z} - \boldsymbol{x}_j) d\boldsymbol{\mu}_{\mathbb{S}^{D-1}} + \sum_{j}^{M} \int_{\mathbf{z} \in \mathbb{S}^{D-1}} |\mathbf{z}^\top \mathbf{b}_i| \delta(\mathbf{z} - \boldsymbol{o}_j) d\boldsymbol{\mu}_{\mathbb{S}^{D-1}} \right)$$

$$= \frac{1}{M+N} \sum_{i=1}^{c'} \left( \int_{\mathbf{z} \in \mathbb{S}^{D-1}} |\mathbf{z}^\top \mathbf{b}_i| \sum_{j=1}^{N} \delta(\mathbf{z} - \boldsymbol{x}_j) d\boldsymbol{\mu}_{\mathbb{S}^{D-1}} + \int_{\mathbf{z} \in \mathbb{S}^{D-1}} |\mathbf{z}^\top \mathbf{b}_i| \sum_{j}^{M} \delta(\mathbf{z} - \boldsymbol{o}_j) d\boldsymbol{\mu}_{\mathbb{S}^{D-1}} \right)$$

$$= \sum_{i=1}^{c'} \left( p \mathbb{E}_{\boldsymbol{\mu}_{\boldsymbol{\mathcal{X}}}}[f_{\mathbf{b}_i}] + (1-p) \mathbb{E}_{\boldsymbol{\mu}_{\boldsymbol{\mathcal{O}}}}[f_{\mathbf{b}_i}] \right)$$

Note that $\boldsymbol{\mu}_{\boldsymbol{\mathcal{X}}}, \boldsymbol{\mu}_{\boldsymbol{\mathcal{O}}}$ arise by discretizing the continuous uniform measures $\boldsymbol{\mu}_{\mathbb{S}^{D-1}}$ and $\boldsymbol{\mu}_{\mathbb{S}^{D-1} \cap \boldsymbol{S}}$ respectively ($\boldsymbol{\mu}_{\mathbb{S}^{D-1} \cap \boldsymbol{S}}$ denotes the uniform measure on $\mathbb{S}^{D-1} \cap \mathcal{S}$) and $p$ is the probability of occurrence of an outlier i.e., $\frac{M}{M+N} \to p$ as $M, N \to \infty$ ( $1 - p$ corresponds to the probability of occurrence of an inlier). That being said, the continuous version of DPCP can be simply stated by replacing $\boldsymbol{\mu}_{\boldsymbol{\mathcal{X}}}, \boldsymbol{\mu}_{\boldsymbol{\mathcal{O}}}$ with $\boldsymbol{\mu}_{\mathbb{S}^{D-1}}$ and $\boldsymbol{\mu}_{\mathbb{S}^{D-1} \cap \boldsymbol{S}}$ in equation 27 as follows,

$$\min_{\mathbf{B}} \sum_{i=1}^{c'} \left( p \mathbb{E}_{\boldsymbol{\mu}_{\mathbb{S}^{D-1}}}[f_{\mathbf{b}_i}] + (1-p) \mathbb{E}_{\boldsymbol{\mu}_{\mathbb{S}^{D-1} \cap \boldsymbol{S}}}[f_{\mathbf{b}_i}] \right) \tag{27}$$

The RHS of (9) immediately shows up by invoking Proposition 4 in Tsakiris & Vidal (2018). ∎

## A.2 PROOF OF LEMMA 3

**Lemma 3**: A projected subgradient algorithm consisting of the steps described in (10) using a piecewise geometrically diminishing step size rule (see (8) in Theorem 1) will almost surely asymptotically converge to a matrix $\hat{\boldsymbol{B}}^* \in \mathbb{R}^{D \times c'}$ whose columns $\hat{\boldsymbol{b}}_i^*$, $i = 1, 2, \ldots, c'$ will be normal vectors of the inliers' subspace when randomly initialized with vectors $\boldsymbol{b}_i^0 \in \mathbb{S}^{D-1}$, $i = 1, 2, \ldots, c'$ uniformly distributed over the sphere $\mathbb{S}^{D-1}$.
**Proof**:

The proof can be trivially obtained by noticing a) that the condition for convergence i.e., inequality (7) of the projected subgradient algorithm given in Theorem 1 becomes $\theta_i^0 < \frac{\pi}{2}$ in the continuous case (since $\eta_{\boldsymbol{X}} \to 0$, $\eta_{\boldsymbol{O}} \to 0$, $c_{\boldsymbol{X},\min} \to c_d > 0$) and b) the set of unit $\ell_2$-norm vectors $\boldsymbol{b}_i^0$s, $i = 1, 2, \ldots, c'$ sampled independently by a uniform distribution over the sphere and whose principal angle $\theta_i^0$ is $\frac{\pi}{2}$ form the inliers' subspace has measure 0. ∎

## A.3 PROOF OF THEOREM 5

**Lemma 4** *The PSGM iterates $\hat{\boldsymbol{b}}_i^k$, $i = 1, 2, \ldots, c'$, $k = 1, 2, \ldots$ given in (10), when randomly initialized with $\hat{\boldsymbol{b}}_i^0$s, $i = 1, 2, \ldots, c'$ that are independently drawn from a spherical distribution with unit $\ell_2$ norm converge almost surely to $c'$ normal vectors of the inliers subspace $\mathcal{S}$ denoted as $\hat{\boldsymbol{b}}_i^*$, $i = 1, 2, \ldots, c'$ that are given by*

$$\hat{\boldsymbol{b}}_i^* = \frac{\mathcal{P}_{\mathcal{S}_\perp}(\hat{\boldsymbol{b}}_i^0)}{\|\mathcal{P}_{\mathcal{S}_\perp}(\hat{\boldsymbol{b}}_i^0)\|_2}, \quad i = 1, 2, \ldots, c' \tag{28}$$

**Proof** Let us assume $\boldsymbol{b}_i^0 = \hat{\boldsymbol{b}}_i^0$. The iterates of subgradients steps of PSGM can be written in the following form,

$$
\begin{aligned}
\boldsymbol{b}_i^1 &= (1 - \mu_i^0 p c_D)\hat{\boldsymbol{b}}_i^0 - \mu_i^0(1-p)c_d\hat{\mathbf{s}} \\
\boldsymbol{b}_i^2 &= (1 - \mu_i^1 p c_D)\hat{\boldsymbol{b}}_i^1 - \mu_i^1(1-p)c_d\hat{\mathbf{s}} \\
\vdots\; &= \qquad\quad \vdots \\
\boldsymbol{b}_i^K &= (1 - \mu_i^{K-1} p c_D)\hat{\boldsymbol{b}}_i^{K-1} - \mu_i^{K-1}(1-p)c_d\hat{\mathbf{s}}
\end{aligned}
\tag{29}
$$

By projecting each update of PSGM onto $\mathcal{S}_\perp$ and since $\mathcal{P}_{\mathcal{S}_\perp}(\boldsymbol{b}_i^k) = \|\boldsymbol{b}_i^k\|\mathcal{P}_{\mathcal{S}_\perp}(\hat{\boldsymbol{b}}_i^k)$ we have,

$$
\mathcal{P}_{\mathcal{S}_\perp}(\boldsymbol{b}_i^{k+1}) = \frac{(1 - \mu_i^k p c_D)}{\|\boldsymbol{b}_i^k\|}\mathcal{P}_{\mathcal{S}_\perp}(\boldsymbol{b}_i^k)
\tag{30}
$$

We can thus easily derive the following form for $\mathcal{P}_{\mathcal{S}_\perp}(\boldsymbol{b}_i^K)$,

$$
\mathcal{P}_{\mathcal{S}_\perp}(\boldsymbol{b}_i^K) = \left(\prod_{k=1}^{K-1} \frac{(1 - \mu_i^k p c_D)}{\|\boldsymbol{b}_i^k\|_2}\right)\mathcal{P}_{\mathcal{S}_\perp}(\boldsymbol{b}_i^0)
\tag{31}
$$

We know from Theorem 1 and Lemma 3 when DPCP-PSGM is initialized with $\boldsymbol{b}_i^0$, $i = 1, 2, \ldots, c'$s randomly drawn according according to a spherical distribution then it will almost surely converge as $K \to \infty$ to vectors $\hat{\boldsymbol{b}}_i^*$, $i = 1, 2, \ldots, c'$ i.e., $\hat{\boldsymbol{b}}_i^K \to \hat{\boldsymbol{b}}_i^*$ where $\hat{\boldsymbol{b}}_i^* \in \mathcal{S}_\perp$. Hence $\mathcal{P}_{\mathcal{S}_\perp}(\hat{\boldsymbol{b}}_i^K) \to \hat{\boldsymbol{b}}_i^*$ as $K \to \infty$. Note that from Theorem 1 we have that $\mu_i^k \neq \frac{1}{pc_D}$ $\forall k = \{1, 2, \ldots, K\}$ hence $\prod_{k=1}^{K-1} \frac{(1-\mu_i^k pc_D)}{\|\hat{\boldsymbol{b}}_i^k\|_2} \neq 0$. From 31 and after projecting on the unit sphere and we thus have $\hat{\boldsymbol{b}}_i^* = \frac{\mathcal{P}_{\mathcal{S}_\perp}(\hat{\boldsymbol{b}}_i^0)}{\|\mathcal{P}_{\mathcal{S}_\perp}(\hat{\boldsymbol{b}}_i^0)\|_2}$. ∎

**Theorem 5** *Let $\hat{\boldsymbol{B}}^0 \in \mathbb{R}^{D \times c'}$ where $c' \geq c$ with $c$ denoting the true codimension of the inliers subspace $\mathcal{S}$, consisting of unit $\ell_2$ norm column vectors $\hat{\boldsymbol{b}}_i^0 \in \mathbb{S}^{D-1}$, $i = 1, 2, \ldots, c'$ that are independently drawn from uniform distribution over the sphere $\mathbb{S}^{D-1}$. A PSGM algorithm initialized with $\hat{\boldsymbol{B}}^0$ will almost surely converge to a matrix $\hat{\boldsymbol{B}}^*$ such that $\mathrm{span}(\hat{\boldsymbol{B}}^*) \equiv \mathcal{S}_\perp$.*

**Proof** From Lemma 4 we have that for each initial unit norm vector $\boldsymbol{b}_i^0$ which corresponds to the $i$th column of $\boldsymbol{B}^0$ will almost surely converge to $\hat{\boldsymbol{b}}_i^* = \frac{\mathcal{P}_{\mathcal{S}_\perp}(\hat{\boldsymbol{b}}_i^0)}{\|\mathcal{P}_{\mathcal{S}_\perp}(\hat{\boldsymbol{b}}_i^0)\|_2}$. We can thus write $\boldsymbol{B}^* = \mathcal{P}_{\mathcal{S}_\perp}(\boldsymbol{B}^0)\boldsymbol{\Gamma}$ where $\boldsymbol{\Gamma}$ is a full-rank diagonal matrix given be $\boldsymbol{\Gamma} = \mathrm{diag}\left(\frac{1}{\|\mathcal{P}_{\mathcal{S}_\perp}(\hat{\boldsymbol{b}}_1^0)\|_2}, \frac{2}{\|\mathcal{P}_{\mathcal{S}_\perp}(\hat{\boldsymbol{b}}_2^0)\|_2}, \ldots, \frac{1}{\|\mathcal{P}_{\mathcal{S}_\perp}(\hat{\boldsymbol{b}}_{c'}^0)\|_2}\right)$. Note that $\mathcal{P}_{\mathcal{S}_\perp}$ is a linear projection and thus we can write $\mathcal{P}_{\mathcal{S}_\perp}(\boldsymbol{B}^0) = \boldsymbol{B}_{\mathcal{S}_\perp}\boldsymbol{B}_{\mathcal{S}_\perp}^\top$ where $\boldsymbol{B}_{\mathcal{S}_\perp}\mathbb{R}^{D \times c}$ is an orthonormal matrix which spans $\mathcal{S}_\perp$. Note that the probability of sampling a low-rank matrix $\boldsymbol{B}_0 = [\boldsymbol{b}_1^0, \boldsymbol{b}_2^0, \ldots, \boldsymbol{b}_{c'}^0]$ when columns $\boldsymbol{b}_i^0$s are randomly and independently drawn from a spherical distribution is zero. We thus have $\boldsymbol{B}^* = \boldsymbol{B}_{\mathcal{S}_\perp}\boldsymbol{B}_{\mathcal{S}_\perp}^\top\boldsymbol{B}^0\boldsymbol{\Gamma}$ with $\mathrm{rank}(\boldsymbol{B}^*) = c$. ∎

**Lemma 6** *If $\sigma_{c'}(\boldsymbol{B}^0) > \|\hat{\boldsymbol{\Delta}}\|_2$ then the rank of matrix $\boldsymbol{B}^0 - \hat{\boldsymbol{\Delta}}$ equals $c'$.*

**Proof** Let $\boldsymbol{A} = \boldsymbol{B}^0 - \hat{\boldsymbol{\Delta}}$. From singular value perturbation inequalities we have $|\sigma_i(\boldsymbol{A}) - \sigma_i(\boldsymbol{B}^0)| \leq \|\hat{\boldsymbol{\Delta}}\|_2$, for $i = 1, 2, \ldots, c'$. Hence it holds,

$$
-\|\hat{\boldsymbol{\Delta}}\|_2 \leq \sigma_i(\boldsymbol{A}) - \sigma_i(\boldsymbol{B}^0)
\tag{32}
$$

If $\sigma_{c'}(\boldsymbol{B}^0) > \|\hat{\boldsymbol{\Delta}}\|_2$ then from equation 32 we get

$$
\sigma_{c'}(\boldsymbol{A}) > 0
\tag{33}
$$

hence the matrix $\boldsymbol{B}^0 - \hat{\boldsymbol{\Delta}}$ will be full-rank. ∎

### A.4 PROOF OF THEOREM 7

We first give the following Lemmas:

**Lemma 7** *For the $\ell_2$ norm of $e_{\mathcal{O}}^{i,k}$ for any $k = 1, 2, \ldots, K$ and $i = 1, 2, \ldots, c'$ it holds,*

$$\|e_{\mathcal{O}}^{i,k}\|_2 \le \eta_{\mathcal{O}} + c_{\mathcal{O},\max} - c_d \tag{34}$$

**Proof**

$$\begin{aligned}
\|e_{\mathcal{O}}^{i,k}\|_2 &= \|o_b - c_d \mathbf{b}\|_2 = \|\frac{1}{M}\mathcal{O}\mathrm{Sgn}(\mathcal{O}^\top b) - c_d b\|_2 \\
&= \|\frac{1}{M}(\mathbf{I} - bb^\top)\mathcal{O}\mathrm{Sgn}(\mathcal{O}^\top b) + \frac{1}{M}bb^\top\mathcal{O}\mathrm{Sgn}(\mathcal{O}^\top b) - c_d b\|_2 \\
&\le \|\frac{1}{M}(\mathbf{I} - bb^\top)\mathcal{O}\mathrm{Sgn}(\mathcal{O}^\top b)\|_2 + (\frac{1}{M}\|\mathcal{O}^\top b\|_1 - c_d)\|b\|_2 \\
&\le \eta_{\mathcal{O}} + c_{\mathcal{O},\max} - c_d
\end{aligned} \tag{35}$$

where we have used the fact that $\|b\|_2 = 1$. ∎

**Lemma 8** *Let the step size of Algorithm 1 (DPCP-PSGM) $\mu_i^k$ being updated following the piecewise geometrically diminishing step size rule with*

$$\beta < \left( \frac{1 - \mu_i^0 M c_D}{1 + \mu_i^0\left(N(\eta_{\mathcal{X}} + c_{\mathcal{X},\max}) + M(\eta_{\mathcal{O}} + c_{\mathcal{O},\max})\right)} \right)^{K_*}.$$

*For the spectral norm of $\hat{\mathbf{\Delta}}$ it holds $\|\hat{\mathbf{\Delta}}\|_2 \le \sqrt{c'}\kappa(\eta_{\mathcal{O}} + c_{\mathcal{O},\max} - c_d)$ where $\kappa = \max_i \frac{M\mu_i^0}{\beta^{K_0/K_*}(1-r_i)}$ and $r_i = \frac{\left(1+\mu_i^0(N(\eta_{\mathcal{X}}+c_{\mathcal{X},\max})+M(\eta_{\mathcal{O}}+c_{\mathcal{O},\max}))\right)}{1-\mu_i^0 M c_D}\beta^{1/K_*}.$*

**Proof** We first bound the $\ell_2$ norm of vectors $\mathbf{b}_i^j$s. We have that $\forall i = 1, 2, \ldots, c'$ and $j = 1, 2, \ldots, K$ it holds

$$b_i^{j+1} = \hat{b}_i^j - \mu_i^j\left(\mathcal{X}\mathrm{Sgn}(\mathcal{X}^\top\hat{b}_i^j) + \mathcal{O}\mathrm{Sgn}(\mathcal{O}^\top\hat{b}_i^j)\right) \tag{36}$$

We define the quantities

$$\eta_{\mathcal{X}} := \max_{\mathbf{b}\in\mathbb{S}^{D-1}} \frac{1}{N}\|(\mathcal{P}_{\mathcal{S}} - \hat{\mathbf{b}}_i^j\hat{\mathbf{b}}_i^{j,\top})\mathcal{X}\mathrm{Sgn}(\mathcal{X}^\top\hat{\mathbf{b}}_i^j)\|_2 \tag{37}$$

$$c_{\mathcal{X},\max} := \max_{\mathbf{b}\in\mathbb{S}^{D-1}} \frac{1}{N}\|\mathcal{X}^\top\hat{\mathbf{b}}_i^j\|_1 \tag{38}$$

$\|\hat{\mathbf{b}}_i^j\|_2 = 1$ hence

$$\begin{aligned}
\|b_i^{j+1}\|_2 &\le 1 + \mu_i^j\|\mathcal{X}\mathrm{Sgn}(\mathcal{X}^\top\hat{b}_i^j) + \mathcal{O}\mathrm{Sgn}(\mathcal{O}^\top\hat{b}_i^j)\|_2 \\
&\le 1 + \mu_i^j\left(\|\mathcal{X}\mathrm{Sgn}(\mathcal{X}^\top\hat{b}_i^j)\|_2 + \|\mathcal{O}\mathrm{Sgn}(\mathcal{O}^\top\hat{b}_i^j)\|_2\right) \\
&\le 1 + \mu_i^j(\|(\mathcal{P}_{\mathcal{S}} - \hat{b}_i^j\hat{b}_i^{j,\top})\mathcal{X}\mathrm{Sgn}(\mathcal{X}^\top\hat{b}_i^j)\|_2 + \|(\hat{b}_i^j\hat{b}_i^{j,\top})\mathcal{X}\mathrm{Sgn}(\mathcal{X}^\top\hat{b}_i^j)\|_2 \\
&\quad + \|\left(1 - \hat{b}_i^j\hat{b}_i^{j,\top}\right)\mathcal{O}\mathrm{Sgn}(\mathcal{O}^\top\hat{b}_i^j)\|_2 + \|\hat{b}_i^j\hat{b}_i^{j,\top}\mathcal{O}\mathrm{Sgn}(\mathcal{O}^\top\hat{b}_i^j)\|_2) \\
&\le 1 + \mu_i^j\left(N(\eta_{\mathcal{X}} + c_{\mathcal{X},\max}) + M(\eta_{\mathcal{O}} + c_{\mathcal{O},\max})\right)
\end{aligned}$$

Due to equation 39 and since $\mu_i^j$ follows a non-increasing path as $j \to k$, the scalar term $\prod_{j=0}^k \frac{\|\mathbf{b}_i^j\|_2}{(1-\mu_i^j M c_D)}$ is bounded above as follows,

$$\prod_{j=0}^k \frac{\|\mathbf{b}_i^j\|_2}{(1-\mu_i^j M c_D)} \le \left( \frac{1 + \mu_i^0\left(N(\eta_{\mathcal{X}} + c_{\mathcal{X},\max}) + M(\eta_{\mathcal{O}} + c_{\mathcal{O},\max})\right)}{1 - \mu_i^0 M c_D} \right)^k \tag{39}$$

We now focus on the geometrically diminishing step size rule given in equation 8. We have $\mu_i^k = \mu_i^0 \beta^{\lfloor k-K_0/K_* \rfloor +1} < \mu_i^0 \beta^{(k-K_0)/K_*}$ for $k \geq K_0$ and $\mu_i^0 \beta^{(k-K_0)/K_*} > \mu_i^0$ for $k < K_0$. Hence we can get the following upper bound

$$\lim_{K \to \infty} \sum_{k=0}^{K-1} \prod_{j=0}^{k} \frac{\|\boldsymbol{b}_i^j\|_2}{(1-\mu_i^j M c_D)} \mu_i^k M <$$

$$M \lim_{K \to \infty} \sum_{k=0}^{K-1} \left( \frac{\left(1+\mu_i^0 \left(N(\boldsymbol{\eta_X}+c_{\boldsymbol{X},\max}) + M(\boldsymbol{\eta_O}+c_{\boldsymbol{O},\max})\right)\right)}{1-\mu_i^0 M c_D} \right)^k \mu_i^0 \beta^{(k-K_0)/K_*}$$

$$\equiv \lim_{K \to \infty} \sum_{k=0}^{K-1} M \frac{1}{\beta^{K_0/K_*}} \mu_i^0 \left( \frac{\left(1+\mu_i^0 \left(N(\boldsymbol{\eta_X}+c_{\boldsymbol{X},\max}) + M(\boldsymbol{\eta_O}+c_{\boldsymbol{O},\max})\right)\right)}{1-\mu_i^0 M c_D} \beta^{1/K_*} \right)^k$$

The series $S = \sum_{k=0}^{K-1} \left( \frac{\left(1+\mu_i^0(N(\boldsymbol{\eta_X}+c_{\boldsymbol{X},\max})+M(\boldsymbol{\eta_O}+c_{\boldsymbol{O},\max}))\right)}{1-\mu_i^0 M c_D} \beta^{1/K_*} \right)^k$ is geometric and if

$$\beta < \left( \frac{1-\mu_i^0 M c_D}{\left(1+\mu_i^0 \left(N(\boldsymbol{\eta_X}+c_{\boldsymbol{X},\max}) + M(\boldsymbol{\eta_O}+c_{\boldsymbol{O},\max})\right)\right)} \right)^{K_*} \tag{40}$$

it converges as $K \to \infty$ to $\frac{1}{1-r_i}$ where $r_i = \frac{\left(1+\mu_i^0(N(\boldsymbol{\eta_X}+c_{\boldsymbol{X},\max})+M(\boldsymbol{\eta_O}+c_{\boldsymbol{O},\max}))\right)}{1-\mu_i^0 M c_D} \beta^{1/K_*}$.

Let us now bound the $\ell_2$ norms of the columns of $\hat{\boldsymbol{\Delta}}$. From Lemma 7 we have $\|\boldsymbol{e}_O^{i,k}\|_2 \leq \boldsymbol{\eta_O} + c_{\boldsymbol{O},\max} - c_d$. We can easily thus derive that $\|\hat{\boldsymbol{\delta}}_i\|_2 \leq \kappa(\boldsymbol{\eta_O} + c_{\boldsymbol{O},\max} - c_d)$. For the spectral norm of $\hat{\boldsymbol{\Delta}}$ we thus have

$$\|\hat{\boldsymbol{\Delta}}\|_2 = \sup_{\boldsymbol{x} \neq \boldsymbol{0}} \frac{\|\hat{\boldsymbol{\Delta}}\boldsymbol{x}\|_2}{\|\boldsymbol{x}\|_2} = \sup_{\boldsymbol{x} \neq \boldsymbol{0}} \frac{\|\sum_{i=1}^{c'} \hat{\boldsymbol{\delta}}_i x_i\|_2}{\|\boldsymbol{x}\|_2} \leq \sup_{\boldsymbol{x} \neq \boldsymbol{0}} \frac{\sum_{i=1}^{c'} \|\hat{\boldsymbol{\delta}}_i\|_2 |x_i|}{\|\boldsymbol{x}\|_2}$$

$$\leq \max_i \|\hat{\boldsymbol{\delta}}_i\|_2 \sup_{\boldsymbol{x} \neq \boldsymbol{0}} \frac{\|\boldsymbol{x}\|_1}{\|\boldsymbol{x}\|_2} \leq \kappa(\boldsymbol{\eta_O} + c_{\boldsymbol{O},\max} - c_d)\sqrt{c'} \tag{41}$$

Where the last inequality arises since $\|\boldsymbol{x}\|_1 \leq \sqrt{c'}\|\boldsymbol{x}\|_2$. $\blacksquare$

We then give the Theorem.

**Theorem 9** *(Theorem 5.58 of Vershynin (2010)) Let $\boldsymbol{B}$ be a $D \times d$ matrix ($D \geq d$) whose columns $\boldsymbol{b}_i$ are independent sub-gaussian isotropic random vectors in $\mathbb{R}^D$ with $\|\boldsymbol{b}_i\|_2 = \sqrt{D}$ almost surely. Then for every $t \geq 0$ the inequality*

$$\sqrt{D} - C\sqrt{d} - t \leq \sigma_{\min}(\boldsymbol{B}) \leq \sigma_{\max}(\boldsymbol{B}) \leq \sqrt{D} + C\sqrt{d} + t \tag{42}$$

*with probability at least $1-2\exp(-ct^2)$, where $C = C'_k, c = c'_K > 0$ depend only on the subgaussian norm $K = \max_j \|\boldsymbol{b}_i\|_{\psi_2}$ of the columns.*

The proof of Theorem 7 follows next.

**Theorem 7** *Let $\mathbf{B}^0 \in \mathbb{R}^{D \times c'}$ with columns randomly sampled from a unit $\ell_2$ norm spherical distribution where $c' \geq c$ with $c$ denoting the true codimension of the inliers subspace $\mathcal{S}$ that satisfies Assumption 1. If*

$$1 - C_1 \sqrt{\frac{c'}{D}} - \frac{\epsilon}{\sqrt{D}} > \sqrt{c'}\kappa(\boldsymbol{\eta_O} + c_{\boldsymbol{O},\max} - c_d) \tag{43}$$

*where $\kappa = \max_i \frac{M\mu_i^0}{\beta^{K_0/K_*}(1-r_i)}$ and $r_i = \frac{\left(1+\mu_i^0(N(\boldsymbol{\eta_X}+c_{\boldsymbol{X},\max})+M(\boldsymbol{\eta_O}+c_{\boldsymbol{O},\max}))\right)}{1-\mu_i^0 M c_D} \beta^{1/K_*}$ then with probability at least $1 - 2\exp(-\epsilon^2 C_2)$ (where $C_1, C_2$ are absolute constants), Algorithm 1 with a geometrically diminishing step size rule will converge to a matrix $\hat{\boldsymbol{B}}^*$ such that $\mathrm{span}(\hat{\boldsymbol{B}}^*) \equiv \mathcal{S}_\perp$.*

Table 1: Results on Washinghton DC AVIRIS hyperspectral image

| Methods | F1-scores | |
|---|---|---|
| | r = 80% | r = 90% |
| DPCP-PSGM (unknown $c$) | 0.994 | 0.993 |
| RSGM (unknown $c$) | 0 | 0 |
| DPCP-IRLS (unknown $c$) | 0 | 0 |
| RSGM ($c = 5$) | 0.999 | 0.993 |
| DPCP-IRLS $c = 5$ | 1 | 0.995 |

**Proof** By Assumption 1 we have that all columns of $\boldsymbol{B}_0$ will satisfy the sufficient condition for convergence of DPCP-PSGM (Algorithm 1) to a normal vector of $\mathcal{S}$. From Lemma 6 and we use the inequality $\sigma_{c'}(\boldsymbol{B}_0) > \|\hat{\boldsymbol{\Delta}}\|_2$ which ensures full-rankness of $\hat{\boldsymbol{B}}^*$, which is the key ingredient in order to prove that $\mathrm{span}(\hat{\boldsymbol{B}}^*) = \mathcal{S}_\perp$. We can then Use Theorem 9 for matrix $\boldsymbol{B}_0$. Note that columns of $\boldsymbol{B}_0$ are drawn independently and are uniformly distributed on the unit sphere. Hence, columns of $\boldsymbol{B}_0$ are sampled by subgaussian distribution and the LHS of the inequality of the theorem appears if we scale with $\frac{1}{\sqrt{D}}$ so that to create unit-norm columns and use LHS of the inequality of Theorem 7. The RHS of the inequality is due to the upper bound of $\|\hat{\boldsymbol{\Delta}}\|_2$ as stated in Lemma 8. The absolute constants $C_1, C_2$ depend only the subgaussian norm of the uniform distribution (they is no dependency on the dimensions of the problem). ∎

## B EXPERIMENTAL DETAILS AND ADDITIONAL MATERIAL

All experiments were conducted on a MacBook Pro 2.6GhZ 6-Core Intel Core i7, memory 16GB 2667 Mhz DDR using Matlab2019B. For computational purposes and in order to avoid fine-tuning of the piecewise geometrically diminishing (PGD) step size, the modified backtracking line-search (MBLS) step-size rule was adopted for DPCP-PSGM as proposed in Zhu et al. (2018). We define the distance between two subspaces spanned by matrices $\boldsymbol{B}$ and $\boldsymbol{A}$ as $dist(\boldsymbol{B}, \boldsymbol{A}) = \min_{\boldsymbol{Q} \in \mathbb{O}(D,c)} \|\boldsymbol{B} - \boldsymbol{A}\boldsymbol{Q}\|_F$ where $\mathbb{O}(D, c)$ denotes the Stiefel manifold of orthogonal matrices of rank $c$. Note that $dist(\boldsymbol{B}, \boldsymbol{A}) = 0 \iff \mathrm{span}(\boldsymbol{B}) \equiv \mathrm{span}(\boldsymbol{A})$ (see Zhu et al. (2019)).

### B.1 OUTLIERS PURSUIT IN WASHINGTON DC MALL AVIRIS HSI

Hyperspectral images (HSIs) provide rich spectral information as compared to RGB images capturing a wide range of the electromagnetic spectrum. Washington DC Mall AVIRIS HSI contains contiguous spectral bands captured at 0.4 to 2.4$\mu$m region of visible and infrared spectrum, Giampouras et al. (2019). In this experiments we randomly choose 10 out of its 210 spectral bands. Due to high coherence in the both the spectral and the spatial domain, pixels of HSIs admit representations in low-dimensional subspaces. Here, we use a 100×100 segment of the hyperspectral image selecting randomly 10 out of its $D = 210$ spectral bands. We form a matrix $\tilde{\boldsymbol{\mathcal{X}}}$ of size $10 \times 10000$ whose columns correspond to different points in the 10-dimensional ambient space. Then we corrupt columns of $\tilde{\boldsymbol{\mathcal{X}}}$ by replacing them with outliers that are generated uniformly at random with unit $\ell_2$ norm for two different outliers' ratios i.e., $r = 0.8$ and $r = 0.9$. In the corrupted $\tilde{\boldsymbol{\mathcal{X}}}$, the remaining clear pixels are considered as the inliers. Table 1 displays the F1 scores obtained by DPCP-PSGM, RSGM and DPCP-IRLS algorithm. The latter two algorithms are evaluated in two scenarios: a) codimension is initialized $c' = 5$ and b) $c' = 10$. Given the singular value distribution of the initial image, we infer that the dimension $d$ of the inliers' subspace is less or equal than 5.

Hence, $c' = 5$ (recall $c = D - d$) is close to the true codimension value while $c' = 10$ is an overestimate thereof. From Table 1, we can see that the proposed DPCP-PSGM succeeds in both outliers' ratios regardless its unawareness of the true codimension value. On the other hand, DPCP-IRLS and RSGM fail when initialized with $c = 10$ and this is attributed to the restrictions induced

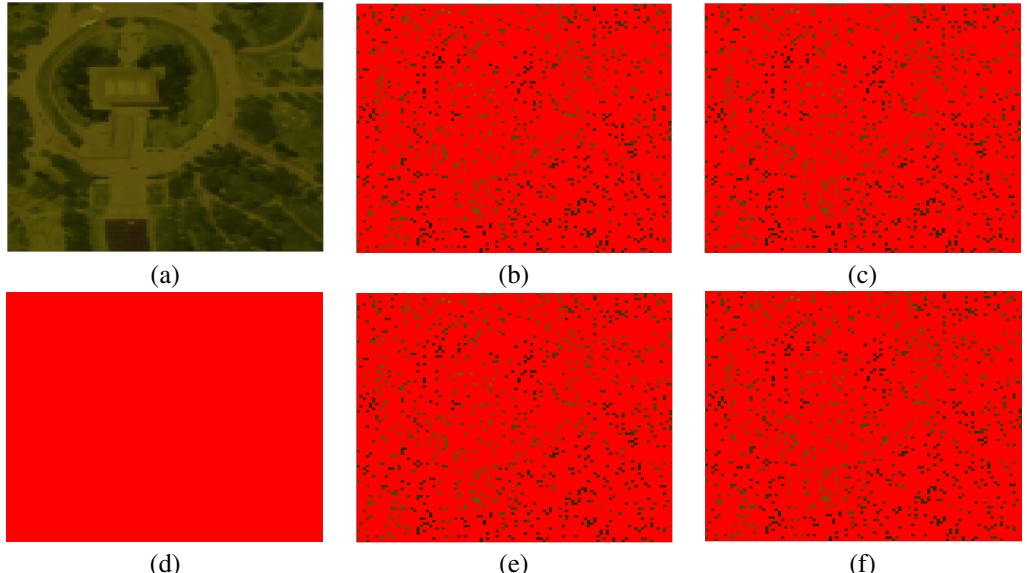

Figure 4: (a) False RGB color image of the clean version of Washington Mall AVIRIS HSI, (b) corrupted by outliers depicted with red and inliers correpsonding the non-red pixels (c) annotated outliers as recoverd by the proposed DPCP-PSGM method initialized with $c' = 10$ (d) RSGM with $c' = 10$, (e) RSGM with $c' = 5$ and (f) DPCP-IRLS with $c' = 5$.

due to the orthogonality constraints they both impose. In Fig. 4 we provide annotated versions of the clean HSI, its corrupted by outliers version for outliers' ratio $r = 90\%$, and the annotated outliers as recovered by the proposed DPCP-PSGM, RSGM, RSGM with $c' = 5$ and DPCP-IRLS with $c' = 5$.

