# OpenReview forum: "Implicit Bias of Projected Subgradient Method Gives Provable Robust Recovery of Subspaces of Unknown Codimension"
_ICLR.cc/2022/Conference — ICLR 2022 Spotlight_

### Official Review · Reviewer_MxBa · 2021-10-27

**Correctness:** 4
**Technical Novelty And Significance:** 3
**Empirical Novelty And Significance:** 3
**Recommendation:** 8
**Confidence:** 2

**Main Review:**

The idea is interesting, and seems to be more of a statistical result than of optimization.  The numerical results are encouraging, and I can't directly find any issues with the proofs (though for time reasons, I did not look too closely).

There are two assumptions I think should be elaborated, at least qualitatively: first, the condition (9) on the initialization of b's, and second, the overestimate of c' over c. These relationships would really give better intuition as to how powerful the random initialization is. Additionally, I am curious as to how much it matters that the data itself has isometry principles. I would suspect if the inlier dimensions have a very uneven contribution, this method should not work that well; how is this captured?

**Summary Of The Paper:**

This paper discusses solving robust subspace recovery by solving parallel versions of $\min_b$ $\|X^Tb\|_1$ st $\| b\|_2=1$, for random initializations of $b$, using projected subgradient method. The argument is that using this  method, in the limit where the number of inliers greatly outnumber the outliers, that if the inlier dimension is c, then c random initializations will converge on identifying this dimension with probability 1. Experimental results and proofs are given to prove this idea.



**Summary Of The Review:**

Overall, I think the paper's idea is nice and looks sound; however, I am not an expert in this area.

---

> ### Author Response · Authors · 2021-11-19
> **Response to Reviewer MxBa**
>
> We would like to thank the reviewer for the positive evaluation of our work.
> We agree with the reviewer that condition in (9) and the overestimate of the subspace codimension are critical for the proposed  PSGM algorithm to converge to a matrix $\hat{\mathbf{B}}$ that will span the correct orthogonal complement of the inliers' subspace. The probability that condition (9) is satisfied depends on the ratio between outliers and inliers and the quantities $\eta_{\mathbf{X}},c_{\mathbf{X},min},\eta_{\mathbf{O}}$ that reflect how well distributed the inliers and outliers in the inliers' subspace and ambient space, respectively. For instance, in the case that inliers and outliers are sampled under continuous measures and their number goes to infinity, condition (9) is satisfied almost surely for all random initial vectors $\mathbf{b}_{i}, i=1,2,\dots,c'$.
>
> Notably, the results reported in the Experimental section show that condition (9) is satisfied even in the more realistic case when a finite number of inliers and outliers is sampled  by discrete distributions. Indeed, as the reviewer points out, the more non-uniformly distributed the inliers and outliers are, the more difficult the task of subspace recovery becomes. This actually can be deduced from the fact that in such cases (when the points are not even distributed) the quantities $\eta_{\mathbf{O}}$ and $\eta_{\mathbf{X}}$ increase hence from (9) we can see that $\theta_{0}$ decreases thus reducing the probability of condition (9) being satisfied when the $\mathbf{b}_{i}$ vectors are randomly initialized. Finally, from Theorem 7 it can be seen that there is a trade-off when it comes to overestimating the inliers' subspace codimension $c'$, since a too high value of that would increase the RHS of the inequality in the theorem making it harder to be satisfied.

---

### Official Review · Reviewer_Az9P · 2021-10-31

**Correctness:** 4
**Technical Novelty And Significance:** 2
**Empirical Novelty And Significance:** 2
**Recommendation:** 6
**Confidence:** 4

**Main Review:**

The work proposed a new analysis framework for justifying the robust subspace recovery under the DPCP formulation without requiring apriori knowledge of the subspace co-dimension. The work analyzed both the gradient flow and its projected subgradient counterpart, showing convergence to the target solutions.
The paper is well-organized, and well-presented overall. The work tackles an important challenge to perform robust subspace recovery in a high relative subspace dimension regime without requiring a priori knowledge of the true subspace dimension.
Nonetheless, the reviewer is a little concerned over the novelty of the work. It seems to be borrowing/combining ideas from previous work on DPCP, and implicit bias for low-rank matrix factorization.


**Summary Of The Paper:**

The work proposed a simple framework that allows performing robust subspace recovery without requiring a priori knowledge of the subspace codimension. The proposed approach is based on Dual Principal Component Pursuit (DPCP), which is amenable to handling subspaces of high relative dimensions. Empirical results corroborate the developed theory and showcase the merits of the proposed optimization methods.

**Summary Of The Review:**

The work is well-presented, and tackles a challenge of unknown true subspace dimension via overparametrization and implicit bias of optimization method. The work combines ideas from previous work on DPCP and implicit bias for low-rank matrix recovery. The overall result is novel, while the approaches seem to be a combination of previous methods.

---

> ### Author Response · Authors · 2021-11-19
> **Response to Reviewer Az9P**
>
> We thank the reviewer for the overall positive assessment of our work as well as for valuing the significance of the problem under study in our paper and for finding our results novel.  The main criticism of the reviewer seems to be how our work is distinct from prior work, which we address next.
>
> $\textbf{Relation to Prior Work on DPCP:}$ We would like to emphasize that all previous works on robust subspace recovery assume that the dimension of the subspace is known. Note also that theoretical results  of previous work  focused on a) the analysis of convergence of problem (1) corresponding to subspaces of codimension equal to 1 b) analysis of  rate of convergence of the PSGM for subspace of codimension equal to 1 and c) analysis of convergence and rate of convergence of a Riemannian subgradient method that minimized a DPCP formulation that incorporated orthogonality constraints and thus assumed the knowledge of the correct subspace codimension.
>
> That being said, none of the existing works addressed the issue of robust subspace recovery when the subspace dimension is not known $\textit{a priori}$, and in the current work we show that all previous DPCP approaches fail to recover the correct subspace when the true subspace dimension is not known precisely.
>
> While our results certainly build on DPCP ideas and methods we note that the approach we propose and analyze is quite unique from previous methods.  Namely, as mentioned above, prior methods assume knowledge of the subspace dimension and require that the optimization iterates maintain strict othogonality constraints.  Our work proves that this is unnecessary.  This has advantages both in the simplicity of the algorithm (simply run the algorithm with no orthogonality constraints) and in provably recovering the $\textit{unknown}$ subspace dimension.  At first, one has no reason to believe that such a simple approach should be expected to succeed, but our analysis shows that this approach will provably recover the correct solution and requires a number of novel theoretical results and proof techniques.
>
> $\textbf{Relation to Prior Work on Implicit Regularization:}$ In terms of the relationship with implicit regularization in low-rank models, at a high level our results are in a similar spirit as this prior work in the sense that they provide guarantees of recovering the correct solution just from the properties of the initialization and optimization dynamics without adding explicit regularization to the model (also see our reply to reviewer [euxB]).  However, our results are completely independent from this work and do not rely on or use any of the existing work from this field.  In fact, we note that our problem is in a rather distinct regime from this work.  Namely, in this prior work the models were highly overparameterized, so there are infinitely many globally optimal solutions and the question is which particular globally optimal solution will be recovered.  In the DPCP problem in (1) there is often a single unique global minimum in terms of simply minimizing the optimization objective, but there a numerous local minimia of (1) which are all orthogonal to the inlier subspace (which is our overall goal).  Here our results actually rely on converging to a diverse set of $\textit{local}$ minima of (1) to recover the correct overall solution (a full basis for the null space of the subspace), which is radically different from the analysis regime typically considered in other works on implicit regularization where there are many non-unique global minima and the question is simply which one is recovered.  We have added a paragraph in the revised version which discusses this prior work on implicit regularization.

---

### Official Review · Reviewer_1qf1 · 2021-11-02

**Correctness:** 4
**Technical Novelty And Significance:** 2
**Empirical Novelty And Significance:** 2
**Recommendation:** 5
**Confidence:** 4

**Main Review:**

The paper introduces a new algorithm for a significant problem, which is able to circumvent an important issue with previous work (no knowledge of the subspace needed). For this reason, I think that the paper has a lot of potential. However, I think that in its current state the paper is not yet ready for acceptance.

Most importantly, I feel that Theorem 7, which is the main result of this paper, needs to be formulated much more cleanly. For example, it is totally unclear to me what is $\beta$. What is $K_{\star}$ and $K_{0}$? Has this been defined before? I feel that the authors should work on a version which is more digestible and more self-contained. Furthermore, I am not able to assess the strength of these results. It might be useful to derive some implications of this theorem in some simple settings to see how sharp the result is.

Furthermore, I think that the paper needs to be really polished and proofread, before it gets submitted again to another conference (see comments below).

Further comments:

1. Figure 1: I think it should be "as opposed to b)" instead of "as opposed to a)"
2. Figure 3: In the caption below the Figure, it is written that both (a) and (b) use r=0.6. I think for (b) it should be r=0.7
3. Is in equation (3) a transpose missing? Should it not be $Xtilde^T$ instead of $Xtilde$?!
4. In equation (7) is $c_{O,min}$ correctly defined? Should it not be $O^T b$ rather than $X^Tb$?
5. This might be a bit nitpicky, but in the related work section on subspace recovery, the authors refer to Joliffe&Cadima and Vidal et al. for PCA, respectively SVD. I feel that if the authors of the paper are doing this they should write "see, e.g., Vidal et. al" instead of just "Vidal et. al". (SVD is not due to Vidal...)
6. I feel that in the related work section on robust subspace recovery some important work is missing. For example Candes, Wright et al. on Robust PCA at the end of page 6: "outliers and inliers terms of (??)
7. In the formulation of Theorem 1 it would be helpful for the reader if the authors would refer to (1). I think that this would clarify that this is the problem which gets solved.
8. Typo in the formulation of Lemma 6: "then (the) matrix"
9. Why do M and N not matter interested formulation of Theorem? Clarifying this might be helpful for the reader.

**Summary Of The Paper:**

This paper is concerned with the Robust Subspace Recovery (RSR) problem. That is, one tries to estimate an unknown subspace given some data, where some of the entries are corrupted. This paper studies the setting, where the subspace to be recovered has large dimension, i.e. the co-dimension of this subspace is small.

Whereas previous work tackles this question, it requires that the dimension of the subspace to be recovered is known a-priori. This work proposes a new algorithm, which is able to recover the subspace without knowing this dimension.

The contributions of this paper are threefold.
1. It proposes a new algorithm, which does not need a-priori knowledge of the unknown subspace.
2. Some theory for the new algorithm is developed.
3. The algorithm is tested on both synthetic and real data.

**Summary Of The Review:**

Due to the aforementioned reasons, I think that the paper is not yet ready for acceptance. However, I feel that the paper has a lot of potential and I hope that the authors keep working on the issues which have been raised in this report.

---

> ### Author Response · Authors · 2021-11-19
> **Response to Reviewer 1qf1**
>
> We thank the reviewer for their positive evaluation of the ``great potential'' of this work. We should note that the quantities $\beta,K_{0},K_{\ast}$ used in Theorem 7, first appear in Theorem 1 given in Section 2 of the paper. Namely, these quantities are related to the piecewise geometrically diminishing step size rule that is followed by the projected subgradient method (PSGM). In the revised version we have added some further explanation/intuition of Theorem 7. We would also like to emphasize that the main objective of simulated data experiments presented in Section 5 is to validate in practice the theoretical result reported in Theorem 7. In these experiments, a finite number of inliers and outliers is sampled uniformly at random. We observe that the proposed approach succeeds in recovering the correct complement of inliers' subspace and the true subspace codimension for a varying number of outliers to inliers ratio and a wide range of true codimension (Figures 2 and 3) thus corroborating the derived theory and showing the efficiency of the proposed DPCP-PSGM algorithm on simulated data.
>
> Minor comments:
>
> - We thank the reviewer for pointing out these typos/errors. We have fixed all of the various typos and minor errors.
> - The reference of the RPCA paper of Candes, et.al. is added.  We also note that RPCA considers a slightly different problem as it considers entries in the matrix being corrupted uniformly at random, whereas here we assume entire data points are corrupted (i.e., corrupting individual entries of a matrix vs corrupting entire columns of a matrix).
> - We are not entirely sure what the reviewer is referring to in comment 9, but we note that in the revised version we have slightly modified the result to be notated in terms of the probability of a point being an outlier or and inlier rather than use the $(M,N)$ notation.

---

### Official Review · Reviewer_euxB · 2021-11-02

**Correctness:** 3
**Technical Novelty And Significance:** 4
**Empirical Novelty And Significance:** 3
**Recommendation:** 8
**Confidence:** 3

**Main Review:**

The main result of this paper is rather surprising-- that random initialization can replace the need for explicit orthogonality constraint in DPCP. These proofs and theoretical results need more careful review, but if this is accurate, I believe it will have meaningful impact on many other problems in signal processing and machine learning.

This paper could be improved by addressing the following points:
- Although the phrase "implicit bias" is in the title and repeated throughout the paper, it is not very well-defined. There should also be a mention on how this result connects to existing works on implicit bias, often mentioned in the context of non-convex optimization.
- Figure 1 needs a lot more explanation in the text and/or the caption. What are the definition of these terms and what is the difference we're supposed to notice between them?
- Sections 3 and 4 are very dense. Is there a way to provide more high level explanations or proof sketches?
- Algorithm 1 is lacking, given the importance of initialization. There should be more explicit instructions on how to find $\hat{B}_0$ inside the algorithm box.
- Are there any computational analysis of the algorithm?


More minor points:
- Should $\tilde{X}$ equation (3) be transposed?
- Notation in equation (6) is confusing. Is the left side claiming that $c_d$ evaluates to $2/\pi$ for even $d$?
- Broken link (??) in page 6
- Figure 2 needs a colorbar. Is this in log or linear scale?

**Summary Of The Paper:**

This paper studies robust sparse recovery when the dimension of the subspace is not known. It finds a basis of the orthogonal complement of the subspace via dual principal component pursuit (DPCP), but without explicitly imposing orthogonality between the columns. The authors propose a projected sub-gradient method with random initialization and overestimate of the dimension, and show theoretical results. Simulation experiments are provided.

**Summary Of The Review:**

The take-home message of the main theoretical result is very interesting, and could open a new way to think about many other optimization problems with explicit orthogonality contrainsts. However, I have only skimmed the proofs and have not carefully checked them.

---

> ### Author Response · Authors · 2021-11-19
> **Response to Reviewer euxB**
>
> We would like to thank the reviewer for their interest in our results and for appreciating the contribution and impact of our work. Next, we address all the reviewer's comments/concerns:
>
> - The term `implicit bias', while not formally defined, often refers to how the choice of optimization algorithm influences the final solution that is recovered, particularly when models are overparameterized with multiple potential solutions.  Here our results are in a similar spirit, in the sense that we prove that the projected subgradient method (PSGM), when started from multiple distinct initializations gives an ensemble of unique solutions which allows one to recover the true desired solution (i.e., the full orthogonal complement of the subspace without knowing the dimension of the subspace  a priori) without any explicit regularization being added. We have added a paragraph to the revised paper which discusses existing work on implicit bias and how our results a related in spirit but of a  distinct nature (see also response to reviewer [Az9p]).
>
> - We have revised the caption of Figure 1 to better explain the importance of random initialization illustrated in Fig.1a  (as opposed to orthogonal initialization that was used in prior works Fig. 1b) in biasing the projected subgradient algorithm to converge to low-rank solution matrix $\hat{B}=c$, where $c$ is the true codimension of subspace $\mathcal{S}$.
>
> - Following the reviewer's comment we revised Sections 3 and 4 and further detailed the intuition behind the theoretical results presented in these sections.
>
> - Matrix $\mathbf{B}_{0}$ is consists of independent columns randomly sampled from a uniform distribution on the sphere. We added the details in the description of Algorithm 1.
>
> -  Being a projected subgradient descent method, Algorithm 1 is computationally efficient with the hardest step per iteration being the computation of gradient which is in the order of $\mathcal{O}(max(M,N)D^{2})$.
>
> Next we respond to the minor comments:
> - Thanks for noticing. We fixed the typo.
> - Yes, $c_{d} = \frac{2}{\pi}$ for even $d$. We improved the format of the equation in the  revised version of the theorem.
> - We fixed the broken link.
> - It is linear scale (number of outliers on the y-axis vs number of inliers on the x-axis). Darker colors correspond to failure and lighter colors to successful recovery.

---

### Author Response · Authors · 2021-11-19
**General Response - Summary of Changes in the Revised Submission**

We would like to thank the reviewers for recognizing the novelty and significance of our work along with finding our main theoretical results `very interesting'. We are also thankful for all of the constructive comments that helped us improve the initial submission. Next, we summarize the major changes we have made in the revised version of the paper.

   1)  We have refined the abstract of the paper.
   2) We have improved the Contributions part in Section 1 in order to better emphasize the intuition of our approach.
   3) In Section 2, we have added a paragraph that elaborates on the use of the term ``implicit bias'' giving a clear insight as to how our approach relates to relevant works in the literature.
   4) We have revised Lemma 2, which presents the continuous version of DPCP problem, in an effort to make it more rigorous. Specifically, the absolute numbers of outliers $M$ and inliers $N$, which both tend to $\infty$ in the continuous case, are now replaced by probabilities of occurrence of an outlier (denoted as $p$) and an inlier ($1-p$), respectively.
  5)  We have revised parts of Section 4 to improve the clarity of the presentation and give a better intuition of Theorem 7.
  6) We have fixed typos/minor mistakes of the first version of the paper that were pointed out by the reviewers.

---

### Decision · Program_Chairs · 2022-01-20

**Decision:**

Accept (Spotlight)

**Comment:**

The paper looks at subspace recovery in the presence of outliers, of which there have been many formulations. They study a recent formulation, DPCP, but relax the requirement that the dimension of the subspace is known -- obviously very important in practice. The approach is quite clever: they exploit the fact that for this non-convex problem, starting a simple algorithm at a randomly chosen starting point will converge to a local minimizer, and they can run an ensemble of these algorithms (each with different starting points) and be guaranteed the solutions will span an appropriate subspace. This idea alone is a nice contribution. The paper has theory and experiments.

Most reviewers were positive about the paper. The most critical review, by 1qf1, still acknowledged that this paper has a lot of potential, but in their opinion the paper was not in a state ready for acceptance, especially regarding the formulation of the main result, Theorem 7.  The other reviewers were OK with the state of the paper, and the authors made changes in the rebuttal. Hence, while acknowledging the paper could possibly still be improved (what paper couldn't be!), I think the paper is in a good enough state to accept it for ICLR. I don't think there would be enough benefit to the community (authors, readers and reviewers) to ask for this to go through one more round of submission/revision.